# Visual Representation Learning Does Not Generalize Strongly Within the Same Domain

**Lukas Schott[1,‡], Julius von Kügelgen[2,3,4], Frederik Träuble[2,4],**
**Peter Gehler[4], Chris Russell[4], Matthias Bethge[1,4], Bernhard Schölkopf[2,4],**
**Francesco Locatello[4,†], Wieland Brendel[1,†]**
[1]University of Tübingen, [2]Max Planck Institute for Intelligent Systems, Tübingen
[3]University of Cambridge, [4]Amazon Web Services
[†]Joint senior authors, [‡]Work done during an internship at Amazon
`lukas.schott@bethgelab.org`

## Abstract

An important component for generalization in machine learning is to uncover underlying latent factors of variation as well as the mechanism through which each factor acts in the world. In this paper, we test whether 17 unsupervised, weakly supervised, and fully supervised representation learning approaches correctly infer the generative factors of variation in simple datasets (dSprites, Shapes3D, MPI3D) from controlled environments, and on our contributed CelebGlow dataset. In contrast to prior robustness work that introduces novel factors of variation during test time, such as blur or other (un)structured noise, we here recompose, interpolate, or extrapolate only existing factors of variation from the training data set (e.g., small and medium-sized objects during training and large objects during testing). Models that learn the correct mechanism should be able to generalize to this benchmark. In total, we train and test 2000+ models and observe that all of them struggle to learn the underlying mechanism regardless of supervision signal and architectural bias. Moreover, the generalization capabilities of all tested models drop significantly as we move from artificial datasets towards more realistic real-world datasets. Despite their inability to identify the correct mechanism, the models are quite modular as their ability to infer other in-distribution factors remains fairly stable, providing only a single factor is out-of-distribution. These results point to an important yet understudied problem of learning mechanistic models of observations that can facilitate generalization.

## 1 Introduction

Humans excel at learning underlying physical mechanisms or inner workings of a system from observations (Funke et al., 2021; Barrett et al., 2018; Santoro et al., 2017; Villalobos et al., 2020; Spelke, 1990), which helps them generalize quickly to new situations and to learn efficiently from little data (Battaglia et al., 2013; Dehaene, 2020; Lake et al., 2017; Téglás et al., 2011). In contrast, machine learning systems typically require large amounts of curated data and still mostly fail to generalize to out-of-distribution (OOD) scenarios (Schölkopf et al., 2021; Hendrycks & Dietterich, 2019; Karahan et al., 2016; Michaelis et al., 2019; Roy et al., 2018; Azulay & Weiss, 2019; Barbu et al., 2019). It has been hypothesized that this failure of machine learning systems is due to shortcut learning (Kilbertus* et al., 2018; Ilyas et al., 2019; Geirhos et al., 2020; Schölkopf et al., 2021). In essence, machines seemingly learn to solve the tasks they have been trained on using auxiliary and spurious statistical relationships in the data, rather than true mechanistic relationships. Pragmatically, models relying on statistical relationships tend to fail if tested outside their training distribution, while models relying on (approximately) the true underlying mechanisms tend to generalize well to novel scenarios (Barrett et al., 2018; Funke et al., 2021; Wu et al., 2019; Zhang et al., 2018; Parascandolo et al., 2018; Schölkopf et al., 2021; Locatello et al., 2020a;b). To learn effective statistical relationships, the training data needs to cover most combinations of factors of variation (like shape, size, color, viewpoint, etc.). Unfortunately, the number of combinations scales exponentially with the number of factors. In contrast, learning the underlying mechanisms behind the factors of variation should greatly reduce the need for training data and scale more gently with the number of factors (Schölkopf et al., 2021; Peters et al., 2017; Besserve et al., 2021).

**Benchmark:** Our goal is to quantify how well machine learning models already learn the mechanisms underlying a data generative process. To this end, we consider four image data sets where each image is described by a small number of independently controllable factors of variation such

as scale, color, or size. We split the training and test data such that models that learned the underlying mechanisms should generalize to the test data. More precisely, we propose several systematic out-of-distribution (OOD) test splits like composition (e.g., *train* = small hearts, large squares → *test* = small squares, large hearts), interpolation (e.g., small hearts, large hearts → medium hearts) and extrapolation (e.g., small hearts, medium hearts → large hearts). While the factors of variation are independently controllable (e.g., there may exist large and small hearts), the observations may exhibit spurious statistical dependencies (e.g., observed hearts are typically small, but size may not be predictive at test time). Based on this setup, we benchmark 17 representation learning approaches and study their inductive biases. The considered approaches stem from un-/weakly supervised disentanglement, supervised learning, and the transfer learning literature.

**Results:** Our benchmark results indicate that the tested models mostly struggle to learn the underlying mechanisms regardless of supervision signal and architecture. As soon as a factor of variation is outside the training distribution, models consistently tend to predict a value in the previously observed range. On the other hand, these models can be fairly modular in the sense that predictions of in-distribution factors remain accurate, which is in part against common criticisms of deep neural networks (Greff et al., 2020; Csordás et al., 2021; Marcus, 2018; Lake & Baroni, 2018).

**New Dataset:** Previous datasets with independent controllable factors such as dSprites, Shapes3D, and MPI3D (Matthey et al., 2017; Kim & Mnih, 2018; Gondal et al., 2019) stem from highly structured environments. For these datasets, common factors of variations are scaling, rotation and simple geometrical shapes. We introduce a dataset derived from celebrity faces, named CelebGlow, with factors of variations such as smiling, age and hair-color. It also contains all possible factor combinations. It is based on latent traversals of a pretrained Glow network provided by Kingma et al. (Kingma & Dhariwal, 2018) and the Celeb-HQ dataset (Liu et al., 2015).

We hope that this benchmark can guide future efforts to find machine learning models capable of understanding the true underlying mechanisms in the data. To this end, all data sets and evaluation scripts are released alongside a leaderboard on GitHub. [1]

## 2 PROBLEM SETTING

Figure 1: Assumed graphical model connecting the *factors of variations* $\mathbf{y} = (y_1, ..., y_n)$ to *observations* $\mathbf{x} = g(\mathbf{y})$. The *selection* variable $s \in \{\text{tr}, \text{te}\}$ leads to different train and test splits $p_s(\mathbf{y})$, thereby inducing correlation between the FoVs.

Assume that we render each observation or image $\mathbf{x} \in \mathbb{R}^d$ using a "computer graphic model" which takes as input a set of independently controllable factors of variation (FoVs) $\mathbf{y} \in \mathbb{R}^n$ like size or color. More formally, we assume a generative process of the form $\mathbf{x} = g(\mathbf{y})$, where $g : \mathbb{R}^n \mapsto \mathbb{R}^d$ is an injective and smooth function. In the standard independently and identically distributed (IID) setting, we would generate the training and test data in the same way, i.e., we would draw $\mathbf{y}$ from the same prior distribution $p(\mathbf{y})$ and then generate the corresponding images $\mathbf{x}$ according to $g(\cdot)$. Instead, we here consider an OOD setting where the prior distribution $p_{\text{tr}}(\mathbf{y})$ during training is different from the prior distribution $p_{\text{te}}(\mathbf{y})$ during testing.

In fact, in all settings of our benchmark, the training and test distributions are completely disjoint, meaning that each point can only have non-zero probability mass in either $p_{\text{tr}}(\mathbf{y})$ or $p_{\text{te}}(\mathbf{y})$. Crucially, however, the function $g$ which maps between FoVs and observations is shared between training and testing, which is why we refer to it as an *invariant mechanism*. As shown in the causal graphical model in Fig. 1, the factors of variations $\mathbf{y}$ are independently controllable to begin with, but the binary split variable $s$ introduces spurious correlations between the FoVs that are different at training and test time as a result of selection bias (Storkey, 2009; Bareinboim & Pearl, 2012). In particular, we consider *Random*, *Composition*, *Interpolation*, and *Extrapolation* splits as illustrated in Fig. 2. We refer to §4.2 for details on the implementation of these splits.

The task for our machine learning models $f$ is to estimate the factors of variations $\mathbf{y}$ that generated the sample $\mathbf{x}$ on both the training and test data. In other words, we want that (ideally) $f = g^{-1}$. The main challenge is that, during training, we only observe data from $p_{\text{tr}}$ but wish to generalize to $p_{\text{te}}$. Hence, the learned function $f$ should not only invert $g$ locally on the training domain $\text{supp}(p_{\text{tr}}(\mathbf{y})) \subseteq \mathbb{R}^n$ but ideally globally. In practice, let $\mathcal{D}_{\text{te}} = \{(\mathbf{y}^k, \mathbf{x}^k)\}$ be the test data with $\mathbf{y}_k$ drawn from $p_{\text{te}}(\mathbf{y})$ and let $f : \mathbb{R}^d \mapsto \mathbb{R}^n$ be the model. Now, the goal is to design and optimize the

---

[1] https://github.com/bethgelab/InDomainGeneralizationBenchmark

Figure 2: **Systematic test and train splits for two factors of variation.** Black dots correspond to the training and red dots to the test distribution. Examples of the corresponding observations are shown on the right.

model $f$ on the training set $\mathcal{D}_{\mathrm{tr}}$ such that it achieves a minimal R-squared distance between $\mathbf{y}^k$ and $f(\mathbf{x}^k)$ on the test set $\mathcal{D}_{\mathrm{te}}$.

During training, models are allowed to sample the data from all non-zero probability regions $\mathrm{supp}(p_{\mathrm{tr}}(\mathbf{y}))$ in whatever way is optimal for its learning algorithm. This general formulation covers different scenarios and learning methods that could prove valuable for learning independent mechanisms. For example, supervised methods will sample an IID data set $\mathcal{D}_{\mathrm{tr}} = \{(\mathbf{y}^k, \mathbf{x}^k)\}$ with $\mathbf{y}^k \sim p_{\mathrm{tr}}(\mathbf{y})$, while self-supervised methods might sample a data set of unlabeled image pairs $\mathcal{D}_{\mathrm{tr}} = \{(\mathbf{x}^k, \tilde{\mathbf{x}}^k)\}$. We aim to understand what inductive biases help on these OOD settings and how to best leverage the training data to learn representations that generalize.

## 3 INDUCTIVE BIASES FOR GENERALIZATION IN VISUAL REPRESENTATION LEARNING

We now explore different types of assumptions, or *inductive biases*, on the representational format (§3.1), architecture (§3.2), and dataset (§3.3) which have been proposed and used in the past to facilitate generalization. Inductive inference and the generalization of empirical findings is a fundamental problem of science that has a long-standing history in many disciplines. Notable examples include Occam's razor, Solomonoff's inductive inference (Solomonoff, 1964), Kolmogorov complexity (Kolmogorov, 1998), the bias-variance-tradeoff (Kohavi et al., 1996; Von Luxburg & Schölkopf, 2011), and the no free lunch theorem (Wolpert, 1996; Wolpert & Macready, 1997). In the context of statistical learning, Vapnik and Chervonenkis (Vapnik & Chervonenkis, 1982; Vapnik, 1995) showed that generalizing from a sample to its population (i.e., IID generalization) requires restricting the capacity of the class of candidate functions—a type of inductive bias. Since shifts between train and test distributions violate the IID assumption, however, statistical learning theory does not directly apply to our types of OOD generalization.

OOD generalization across different (e.g., observational and experimental) conditions also bears connections to causal inference (Pearl, 2009; Peters et al., 2017; Hernán & Robins, 2020). Typically, a causal graph encodes assumptions about the relation between different distributions and is used to decide how to "transport" a learned model (Pearl & Bareinboim, 2011; Pearl et al., 2014; Bareinboim & Pearl, 2016; von Kügelgen et al., 2019). Other approaches aim to learn a model which leads to invariant prediction across multiple environments (Schölkopf et al., 2012; Peters et al., 2016; Heinze-Deml et al., 2018; Rojas-Carulla et al., 2018; Arjovsky et al., 2019; Lu et al., 2021). However, these methods either consider a small number of causally meaningful variables in combination with domain knowledge, or assume access to data from multiple environments. In our setting, on the other hand, we aim to learn from higher-dimensional observations and to generalize from a single training set to a different test environment.

Our work focuses on OOD generalization in the context of visual representation learning, where deep learning has excelled over traditional learning approaches (Krizhevsky et al., 2012; LeCun et al., 2015; Schmidhuber, 2015; Goodfellow et al., 2016). In the following, we therefore concentrate on inductive biases specific to deep neural networks (Goyal & Bengio, 2020) on visual data. For details regarding specific objective functions, architectures, and training, we refer to the supplement.

### 3.1 INDUCTIVE BIAS 1: REPRESENTATIONAL FORMAT

Learning useful representations of high-dimensional data is clearly important for the downstream performance of machine learning models (Bengio et al., 2013). The first type of inductive bias we consider is therefore the *representational format*. A common approach to representation learning is to postulate *independent latent variables* which give rise to the data, and try to infer these in an *unsupervised* fashion. This is the idea behind independent component analysis (ICA) (Comon,

1994; Hyvärinen & Oja, 2000) and has also been studied under the term *disentanglement* (Bengio et al., 2013). Most recent approaches learn a deep generative model based on the variational auto-encoder (VAE) framework (Kingma & Welling, 2013; Rezende et al., 2014), typically by adding regularization terms to the objective which further encourage independence between latents (Higgins et al., 2017; Kim & Mnih, 2018; Chen et al., 2018; Kumar et al., 2018; Burgess et al., 2018).

It is well known that ICA/disentanglement is theoretically non-identifiable without additional assumptions or supervision (Hyvärinen & Pajunen, 1999; Locatello et al., 2018). Recent work has thus focused on *weakly supervised* approaches which can provably identify the true independent latent factors (Hyvärinen & Morioka, 2016; Hyvarinen & Morioka, 2017; Shu et al., 2019; Locatello et al., 2020a; Klindt et al., 2020; Khemakhem et al., 2020; Roeder et al., 2020). The general idea is to leverage additional information in the form of paired observations $(\mathbf{x}^i, \tilde{\mathbf{x}}^i)$ where $\tilde{\mathbf{x}}^i$ is typically an auxiliary variable (e.g., an environment indicator or time-stamp) or a second view, i.e., $\tilde{\mathbf{x}}^i = g(\tilde{\mathbf{y}}^i)$ with $\tilde{\mathbf{y}}^i \sim p(\tilde{\mathbf{y}}|\mathbf{y}^i)$, where $\mathbf{y}^i$ are the FoVs of $\mathbf{x}^i$ and $p(\tilde{\mathbf{y}}|\mathbf{y})$ depends on the method. We remark that such identifiability guarantees only hold for the training distribution (and given infinite data), and thus may break down once we move to a different distribution for testing. In practice, however, we hope that the identifiability of the representation translates to learning mechanisms that generalize.

In our study, we consider the popular $\beta$-VAE (Higgins et al., 2017) as an unsupervised approach, as well as Ada-GVAE (Locatello et al., 2020a), Slow-VAE (Klindt et al., 2020) and PCL (Hyvarinen & Morioka, 2017) as weakly supervised disentanglement methods. First, we learn a representation $\mathbf{z} \in \mathbb{R}^n$ given only (pairs of) observations (i.e., without access to the FoVs) using an encoder $f_{\mathrm{enc}} : \mathbb{R}^d \to \mathbb{R}^n$. We then freeze the encoder (and thus the learned representation $\mathbf{z}$) and train a multi-layer perceptron (MLP) $f_{\mathrm{MLP}} : \mathbb{R}^n \to \mathbb{R}^n$ to predict the FoVs $\mathbf{y}$ from $\mathbf{z}$ in a supervised way. The learned inverse mechanism $f$ in this case is thus given by $f = f_{\mathrm{MLP}} \circ f_{\mathrm{enc}}$.

### 3.2 INDUCTIVE BIAS 2: ARCHITECTURAL (SUPERVISED LEARNING)

The physical world is governed by symmetries (Nother, 1915), and enforcing appropriate task-dependent symmetries in our function class may facilitate more efficient learning and generalization. The second type of inductive bias we consider thus regards properties of the learned regression function, which we refer to as *architectural bias*. Of central importance are the concepts of *invariance* (changes in input should not lead to changes in output) and *equivariance* (changes in input should lead to proportional changes in output). In vision tasks, for example, object *localization* exhibits *equivariance* to translation, whereas object *classification* exhibits *invariance* to translation. E.g., translating an object in an input image should lead to an equal shift in the predicted bounding box (equivariance), but should not affect the predicted object class (invariance).

A famous example is the convolution operation which yields translation equivariance and forms the basis of convolutional neural networks (CNNs) (Le Cun et al., 1989; LeCun et al., 1989). Combined with a set operation such as pooling, CNNs then achieve translation invariance. More recently, the idea of building equivariance properties into neural architectures has also been successfully applied to more general transformations such as rotation and scale (Cohen & Welling, 2016; Cohen et al., 2019; Weiler & Cesa, 2019) or (coordinate) permutations (Zhang et al., 2019; Achlioptas et al., 2018). Other approaches consider affine transformations (Jaderberg et al., 2015), allow to trade off invariance vs dependence on coordinates (Liu et al., 2018), or use residual blocks and skip connections to promote feature re-use and facilitate more efficient gradient computation (He et al., 2016; Huang et al., 2017). While powerful in principle, a key challenge is that relevant equivariances for a given problem may be unknown a priori or hard to enforce architecturally. E.g., 3D rotational equivariance is not easily captured for 2D-projected images, as for the MPI3D data set.

In our study, we consider the following architectures: standard MLPs and CNNs, CoordConv (Liu et al., 2018) and coordinate-based (Sitzmann et al., 2020) nets, Rotationally-Equivariant (Rotation-EQ) CNNs (Cohen & Welling, 2016), Spatial Transformers (STN) (Jaderberg et al., 2015), ResNet (RN) 50 and 101 (He et al., 2016), and DenseNet (Huang et al., 2017). All networks $f$ are trained to directly predict the FoVs $\mathbf{y} \approx f(\mathbf{x})$ in a purely supervised fashion.

### 3.3 INDUCTIVE BIAS 3: LEVERAGING ADDITIONAL DATA (TRANSFER LEARNING)

The physical world is modular: many patterns and structures reoccur across a variety of settings. Thus, the third and final type of inductive bias we consider is leveraging additional data through transfer learning. Especially in vision, it has been found that low-level features such as edges or

simple textures are consistently learned in the first layers of neural networks, which suggests their usefulness across a wide range of tasks (Sun et al., 2017). State-of-the-art approaches therefore often rely on pre-training on enormous image corpora prior to fine-tuning on data from the target task (Kolesnikov et al., 2020; Mahajan et al., 2018; Xie et al., 2020). The guiding intuition is that additional data helps to learn common features and symmetries and thus enables a more efficient use of the (typically small amount of) labeled training data. Leveraging additional data as an inductive bias is connected to the representational format §3.1 as they are often combined during pre-training.

In our study, we consider three pre-trained models: RN-50 and RN-101 pretrained on ImageNet-21k (Deng et al., 2009; Kolesnikov et al., 2020) and a DenseNet pretrained on ImageNet-1k (ILSVRC) (Russakovsky et al., 2015). We replace the last layer with a randomly initialized readout layer chosen to match the dimension of the FoVs of a given dataset and fine-tune the whole network for 50,000 iterations on the respective train splits.

# 4 EXPERIMENTAL SETUP

## 4.1 DATASETS

We consider datasets with images generated from a set of discrete Factors of Variation (FoVs) following a deterministic generative model. All selected datasets are designed such that all possible combinations of factors of variation are realized in a corresponding image. *dSprites* (Matthey et al., 2017), is composed of low resolution binary images of basic shapes with 5 FoVs: shape, scale, orientation, x-position, and y-position. Next, *Shapes3D* (Kim & Mnih, 2018), a popular dataset with 3D shapes in a room with 6 FoVs: floor, color, wall color, object color, object size, object type, and camera azimuth. Furthermore, with *CelebGlow* we introduce a novel dataset that has more natural factors of variations such as smiling, hair-color and age. For more details and samples, we refer to Appendix B.

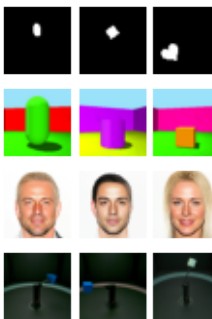

Figure 3: Random dataset samples from dSprites (1st), Shapes3D (2nd), CelebGlow (3rd), and MPI3D-real (4th).

Lastly, we consider the challenging and realistic *MPI3D* (Gondal et al., 2019), which contains real images of physical 3D objects attached to a robotic finger generated with 7 FoVs: color, shape, size, height, background color, x-axis, and y-axis. For more details, we refer to Appendix H.1.

## 4.2 SPLITS

For each of the above datasets, denoted by $\mathcal{D}$, we create disjoint splits of train sets $\mathcal{D}_{\text{tr}}$ and test sets $\mathcal{D}_{\text{te}}$. We systematically construct the splits according to the underlying factors to evaluate different modalities of generalization, which we refer to as *composition*, *interpolation*, *extrapolation*, and *random*. See Fig. 2 for a visual presentation of such splits regarding two factors.

**Composition:** We exclude all images from the train split if factors are located in a particular corner of the FoV hyper cube given by all FoVs. This means certain systematic combinations of FoVs are never seen during training even though the value of each factor is individually present in the train set. The related test split then represents images of which at least two factors resemble such an unseen composition of factor values, thus testing generalization w.r.t. composition.

**Interpolation:** Within the range of values of each FoV, we periodically exclude values from the train split. The corresponding test split then represents images of which at least one factor takes one of the unseen factor values in between, thus testing generalization w.r.t. interpolation.

**Extrapolation:** We exclude all combinations having factors with values above a certain label threshold from the train split. The corresponding test split then represents images with one or more extrapolated factor values, thus testing generalization w.r.t. extrapolation.

**Random:** Lastly, as a baseline to test our models performances across the full dataset in distribution, we cover the case of an IID sampled train and test set split from $\mathcal{D}$. Compared to inter- and extrapolation where factors are systematically excluded, here it is very likely that all individual factor values have been observed in a some combination.

We further control all considered splits and datasets such that $\sim 30\%$ of the available data is in the training set $\mathcal{D}_{\text{tr}}$ and the remaining $\sim 70\%$ belong to the test set $\mathcal{D}_{\text{te}}$. Lastly, we do not split along factors of variation if no intuitive order exists. Therefore, we do not split along the categorical variable *shape* and along the axis of factors where only two values are available.

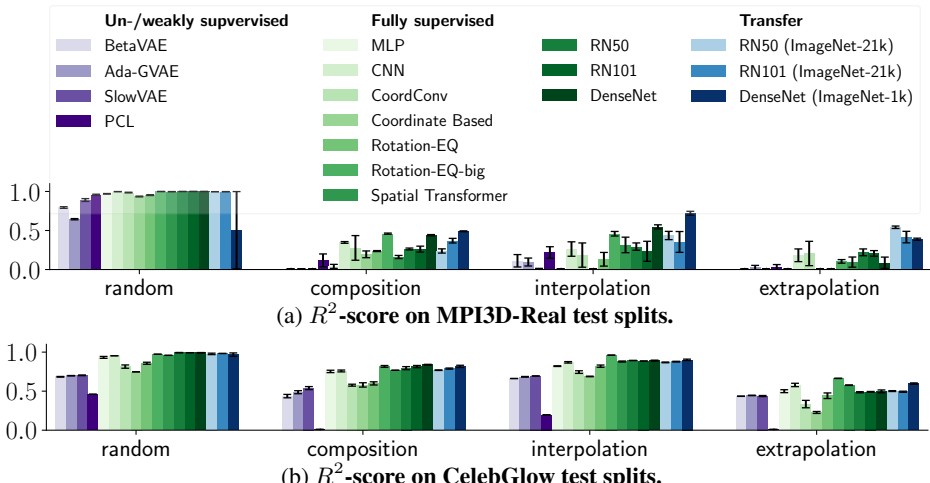

(a) $R^2$-score on MPI3D-Real test splits.

(b) $R^2$-score on CelebGlow test splits.

Figure 4: $R^2$-score on various test-train splits. Compared to the in-distribution random splits, on the out-of-distribution (OOD) splits composition, interpolation, and extrapolation, we observe large drops in performance.

## 4.3 EVALUATION

To benchmark the generalization capabilities, we compute the $R^2$-score, the coefficient of determination, on the respective test set. We define the $R^2$-score based on the MSE score per FoV $y_j$

$$R_i^2 = 1 - \frac{\text{MSE}_i}{\sigma_i^2} \quad \text{with} \quad \text{MSE}_j = \mathbb{E}_{(\mathbf{x},\mathbf{y}) \in D_{\text{te}}} \left[ (\mathbf{y}_j - f_j(\mathbf{x}))^2 \right], \quad (1)$$

where $\sigma_i^2$ is the variance per factor defined on the full dataset $D$. Under this score, $R_i^2 = 1$ can be interpreted as perfect regression and prediction under the respective test set whereas $R_i^2 = 0$ indicates random guessing with the MSE being identical to the variance per factor. For visualization purposes, we clip the $R^2$ to 0 if it is negative. We provide all unclipped values in the Appendix.

## 5 EXPERIMENTS AND RESULTS

Our goal is to investigate how different visual representation models perform on our proposed systematic out-of-distribution (OOD) test sets. We consider un-/weakly supervised, fully supervised, and transfer learning models. We focus our conclusions on MPI3D-Real as it is the most realistic dataset. Further results on dSprites and Shapes3D are, however, mostly consistent.

In the first subsection, §5.1, we investigate the overall model OOD performance. In Sections 5.2 and 5.3, we focus on a more in-depth error analysis by controlling the splits s.t. only a single factor is OOD during testing. Lastly, in §5.4, we investigate the connection between the degree of disentanglement and downstream performance.

## 5.1 MODEL PERFORMANCE DECREASES ON OOD TEST SPLITS

In Fig. 4 and Appendix Fig. 11, we plot the performance of each model across different generalization settings. Compared to the in-distribution (ID) setting (random), we observe large drops in performance when evaluating our OOD test sets on all considered datasets. This effect is most prominent on MPI3D-Real. Here, we further see that, on average, the performances seem to increase as we increase the supervision signal (comparing RN50, RN101, DenseNet with and without additional data on MPI3D). On CelebGlow, models also struggle to extrapolate. However, the results on composition and interpolation only drop slightly compared to the random split.

For Shapes3D (shown in the Appendix E), the OOD generalization is partially successful, especially in the composition and interpolation settings. We hypothesize that this is due to the dataset specific, fixed spatial composition of the images. For instance, with the object-centric positioning, the floor, wall and other factors are mostly at the same position within the images. Thus, they can reliably be inferred by only looking at a certain fixed spot in the image. In contrast, for MPI3D this is more difficult as, e.g., the robot finger has to be found to infer its tip color. Furthermore, the factors of variation in Shapes3D mostly consist of colors which are encoded within the same input dimensions,

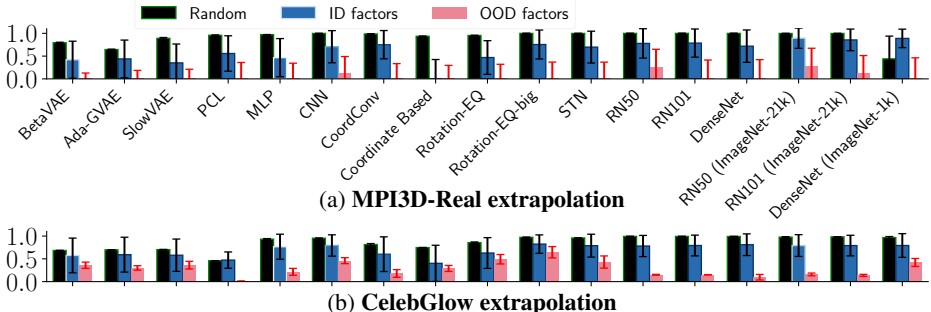

(a) **MPI3D-Real extrapolation**

(b) **CelebGlow extrapolation**

Figure 5: **Extrapolation and modularity, $R^2$-score on subsets.** In the extrapolation setting, we further differentiate between factors that have been observed during training (ID factors) and extrapolated values (OOD factors) and measure the performances separately. As a reference, we compare to a random split. A model is considered modular, if it still infers ID factors correctly despite other factors being OOD.

and not across pixels as, for instance, x-translation in MPI3D. For this color interpolation, the ReLU activation function might be a good inductive bias for generalization. However, it is not sufficient to achieve extrapolations, as we still observe a large drop in performance here.

**Conclusion:** The performance generally decreases when factors are OOD regardless of the supervision signal and architecture. However, we also observed exceptions in Shapes3D where OOD generalization was largely successful except for extrapolation.

### 5.2 Errors stem from inferring OOD factors

While in the previous section we observed a general decrease in $R^2$ score for the interpolation and extrapolation splits, our evaluation does not yet show how errors are distributed among individual factors that are in- and out-of-distribution.

In contrast to the previous section where *multiple* factors could be OOD distribution simultaneously, here, we control data splits (Fig. 2) interpolation, extrapolation s.t. only a *single* factor is OOD. Now, we also estimate the $R^2$-score separately per factor, depending on whether they have individually been observed during training (ID factor) or are exclusively in the test set (OOD factor). For instance, if we only have images of a heart with varying scale and position, we query the model with hearts at larger scales than observed during training (OOD factor), but at a previously observed position (ID factor). For a formal description, see Appendix Appendix H.2. This controlled setup enables us to investigate the modularity of the tested models, as we can separately measure the performance on OOD and ID factors. As a reference for an approximate upper bound, we additionally report the performance of the model on a random train/test split.

In Figs. 5 and 14, we observe significant drops in performance for the OOD factors compared to a random test-train split. In contrast, for the ID factors, we see that the models still perform close to the random split, although with much larger variance. For the interpolation setting (Appendix Fig. 14), this drop is also observed for MPI3D and dSprites but not for Shapes3D. Here, OOD and ID are almost on par with the random split. Note that our notion of modularity is based on systematic splits of individual factors and the resulting outputs. Other works focus on the inner behavior of a model by, e.g., investigating the clustering of neurons within the network (Filan et al., 2021). Preliminary experiments showed no correlations between the different notions of modularity.

**Conclusion:** The tested models can be fairly modular, in the sense that the predictions of ID factors remain accurate. The low OOD performances mainly stem from incorrectly extrapolated or interpolated factors. Given the low inter-/extrapolation (i.e., OOD) performances on MPI3D and dSprites, evidently no model learned to invert the ground-truth generative mechanism.

### 5.3 Models extrapolate similarly and towards the mean

In the previous sections, we observed that our tested models specifically extrapolate poorly on OOD factors. Here, we focus on quantifying the behavior of how different models extrapolate.

To check whether different models make similar errors, we compare the extrapolation behavior across architectures and seeds by measuring the similarity of model predictions for the OOD factors described in the previous section. No model is compared to itself if it has the same random

(a) **MPI3D-Real**          (b) **CelebGlow**          (c) **Shapes3D**          (d) **dSprites**

Figure 6: **Extrapolation towards the mean.** We calculate (2) on the extrapolated OOD factors to measure the closeness towards the mean compared to the ground-truth. Here, the values are mostly in $[0, 1]$. Thus, models tend to predict values in previously observed ranges.

seed. On MPI3D, Shapes3D and dSprites, all models strongly correlate with each other (Pearson $\rho \geq 0.57$) but anti-correlate compared to the ground-truth prediction (Pearson $\rho \leq -0.48$), the overall similarity matrix is shown in Appendix Fig. 17. One notable exception is on CelebGlow. Here, some models show low but positive correlations with the ground truth generative model (Pearson $\rho \geq 0.57$). However, visually the models are still quite off as shown for the model with the highest correlation in Fig. 18. In most cases, the highest similarity is along the diagonal, which demonstrates the influence of the architectural bias. This result hints at all models making similar mistakes extrapolating a factor of variation.

We find that models collectively tend towards predicting the mean for each factor in the training distribution when extrapolating. To show this, we estimate the following ratio of distances

$$r = |f(\mathbf{x}^i)_j - \bar{\mathbf{y}}_j| \, / \, |\mathbf{y}_j^i - \bar{\mathbf{y}}_j|, \tag{2}$$

where $\bar{\mathbf{y}}_j = \frac{1}{n} \sum_{i=1}^n \mathbf{y}_j^i$ is the mean of FoV $y_j$. If values of (2) are $\in [0, 1]$, models predict values which are closer to the mean than the corresponding ground-truth. We show a histogram over all supervised and transfer-based models for each dataset in Fig. 6. Models tend towards predicting the mean as only few values are $>= 1$. This is shown qualitatively in Appendix Figs. 15 and 16.

**Conclusion:** Overall, we observe only small differences in *how* the tested models extrapolate, but a strong difference compared to the ground-truth. Instead of extrapolating, all models regress the OOD factor towards the mean in the training set. We hope that this observation can be considered to develop more diverse future models.

### 5.4    ON THE RELATION BETWEEN DISENTANGLEMENT AND DOWNSTREAM PERFORMANCE

Previous works have focused on the connection between disentanglement and OOD downstream performance (Träuble et al., 2020; Dittadi et al., 2020; Montero et al., 2021). Similarly, for our systematic splits, we measure the degree of disentanglement using the DCI-Disentanglement (Eastwood & Williams, 2018) score on the latent representation of the embedded test and train data. Subsequently, we correlate it with the $R^2$-performance of a supervised readout model which we report in §5.1. Note that the simplicity of the readout function depends on the degree of disentanglement, e.g., for a perfect disentanglement up to permutation and sign flips this would just be an assignment problem. For the disentanglement models, we consider the un-/ weakly supervised models $\beta$-VAE(Higgins et al., 2017), SlowVAE (Klindt et al., 2020), Ada-GVAE(Locatello et al., 2020a) and PCL (Hyvarinen & Morioka, 2017).

We find that the degree of downstream performance correlates positively with the degree of disentanglement (Pearson $\rho = 0.63$, Spearman $\rho = 0.67$). However, the correlations vary per dataset and split (see Appendix Fig. 7). Moreover, the overall performance of the disentanglement models followed by a supervised readout on the OOD split is lower compared to the supervised models (see e.g. Fig. 4). In an ablation study with an oracle embedding that disentangles the test data up to permutations and sign flips, we found perfect generalization capabilities ($R^2_{test} \geq 0.99$).

**Conclusion:** Disentanglement models show no improved performance in OOD generalization. Nevertheless, we observe a mostly positive correlation between the degree of disentanglement and the downstream performance.

## 6    OTHER RELATED BENCHMARK STUDIES

In this section, we focus on related benchmarks and their conclusions. For related work in the context of inductive biases, we refer to §3.

**Corruption benchmarks:** Other current benchmarks focus on the performance of models when adding common corruptions (denoted by -C) such as noise or snow to current dataset test sets,

resulting in ImageNet-C, CIFAR-10-C, Pascal-C, Coco-C, Cityscapes-C and MNIST-C (Hendrycks & Dietterich, 2019; Michaelis et al., 2019; Mu & Gilmer, 2019). In contrast, in our benchmark, we assure that the factors of variations are present in the training set and merely have to be generalized correctly. In addition, our focus lies on identifying the ground truth generative process and its underlying factors. Depending on the task, the requirements for a model are very different. E.g., the ImageNet-C classification benchmark requires spatial invariance, whereas regressing factors such as, e.g., shift and shape of an object, requires in- and equivariance.

**Abstract reasoning:** Model performances on OOD generalizations are also intensively studied from the perspective of abstract reasoning, visual and relational reasoning tasks (Barrett et al., 2018; Wu et al., 2019; Santoro et al., 2017; Villalobos et al., 2020; Zhang et al., 2016; Yan & Zhou, 2017; Funke et al., 2021; Zhang et al., 2018). Most related, (Barrett et al., 2018; Wu et al., 2019) also study similar interpolation and extrapolation regimes. Despite using notably different tasks such as abstract or spatial reasoning, they arrive at similar conclusions: They also observe drops in performance in the generalization regime and that interpolation is, in general, easier than extrapolation, and also hint at the modularity of models using distractor symbols (Barrett et al., 2018). Lastly, posing the concept of using correct generalization as a necessary condition to check whether an underlying mechanism has been learned has also been proposed in (Wu et al., 2019; Zhang et al., 2018; Funke et al., 2021).

**Disentangled representation learning:** Close to our work, Montero et al. (Montero et al., 2021) also study generalization in the context of extrapolation, interpolation and a weak form of composition on dSprites and Shapes3D, but not the more difficult MPI3D-Dataset. They focus on reconstructions of unsupervised disentanglement algorithms and thus the *decoder*, a task known to be theoretically impossible(Locatello et al., 2018). In their setup, they show that OOD generalization is limited. From their work, it remains unclear whether the generalization along known factors is a general problem in visual representation learning, and how neural networks fail to generalize. We try to fill these gaps. Moreover, we focus on representation learning approaches and thus on the *encoder* and consider a broader variety of models, including theoretically identifiable approaches (Ada-GAVE, SlowVAE, PCL), and provide a thorough in-depth analysis of how networks generalize.

Previously, Träuble et al. (2020) studied the behavior of unsupervised disentanglement models on correlated training data. They find that despite disentanglement objectives, the learned latent spaces mirror this correlation structure. In line with our work, the results of their supervised post-hoc regression models on Shapes3D suggest similar generalization performances as we see in our respective disentanglement models in Figs. 4 and 11. OOD generalization w.r.t. extrapolation of one single FoV is also analyzed in (Dittadi et al., 2020). Our experimental setup in §5.4 is similar to their 'OOD2' scenario. Here, our results are in accordance, as we both find that the degree of disentanglement is lightly correlated with the downstream performance.

**Others:** To demonstrate shortcuts in neural networks, Eulig et al. (2021) introduce a benchmark with factors of variations such as color on MNIST that correlate with a specified task but control for those correlations during test-time. In the context of reinforcement learning, Packer et al. (2018) assess models on systematic test-train splits similar to our inter-/extrapolation and show that current models cannot solve this problem. For generative adversarial networks (GANs), it has also been shown that their learned representations do not extrapolate beyond the training data (Jahanian et al., 2019).

# 7 DISCUSSION AND CONCLUSION

In this paper, we highlight the importance of learning the independent underlying mechanisms behind the factors of variation present in the data to achieve generalization. However, we empirically show that among a large variety of models, no tested model succeeds in generalizing to all our proposed OOD settings (extrapolation, interpolation, composition). We conclude that the models are limited in learning the underlying mechanism behind the data and rather rely on strategies that do not generalize well. We further observe that while one factor is out-of-distribution, most other in-distribution factors are inferred correctly. In this sense, the tested models are surprisingly modular.

To further foster research on this intuitively simple, yet unsolved problem, we release our code as a benchmark. This benchmark, which allows various supervision types and systematic controls, should promote more principled approaches and can be seen as a more tractable intermediate milestone towards solving more general OOD benchmarks. In the future, a theoretical treatment identifying further inductive biases of the model and the necessary requirements of the data to solve our proposed benchmark should be further investigated.

ACKNOWLEDGEMENTS

The authors thank Steffen Schneider, Matthias Tangemann and Thomas Brox for their valuable feedback and fruitful discussions. The authors would also like to thank David Klindt, Judy Borowski, Dylan Paiton, Milton Montero and Sudhanshu Mittal for their constructive criticism of the manuscript. The authors thank the International Max Planck Research School for Intelligent Systems (IMPRS-IS) for supporting FT and LS. We acknowledge support from the German Federal Ministry of Education and Research (BMBF) through the Competence Center for Machine Learning (TUE.AI, FKZ 01IS18039A) and the Bernstein Computational Neuroscience Program Tübingen (FKZ: 01GQ1002). WB acknowledges support via his Emmy Noether Research Group funded by the German Science Foundation (DFG) under grant no. BR 6382/1-1 as well as support by Open Philantropy and the Good Ventures Foundation. MB and WB acknowledge funding from the MICrONS program of the Advanced Research Projects Activity (IARPA) via Department of Interior/Interior Business Center (DoI/IBC) contract number D16PC00003.

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

## ETHICS STATEMENT

Our current study focuses on basic research and has no direct application or societal impact. Nevertheless, we think that the broader topic of generalization should be treated with great care. Especially oversimplified generalization and automation without a human in the loop could have drastic consequences in safety critical environments or court rulings.

Large-scale studies require a lot of compute due to multiple random seeds and exponentially growing sets of possible hyperparameter combinations. Following claims by Strubel et al. (Strubell et al., 2019), we tried to avoid redundant computations by orienting ourselves on current common values in the literature and by relying on systematic test runs. In a naive attempt, we tried in to estimate the power consumption and greenhouse gas impact based on the used cloud compute instance. However, too many factors such as external thermal conditions, actual workload, type of power used and others are involved (Mytton, 2020; Fahad et al., 2019). In the future, especially with the trend towards larger network architectures, compute clusters should be required to enable options which report the estimated environmental impact. However, it should be noted that cloud vendors are already among the largest purchasers of renewable electricity (Mytton, 2020).

For an impact statement for the broader field of representation learning, we refer to Klindt et al. (2020).

## REPRODUCIBILITY STATEMENT

Our code is attached, and all important details to reproduce our results are repeated in Appendix H.

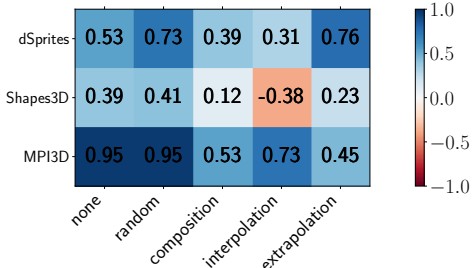

Figure 7: **Spearman Correlation of degree of disentanglement with downstream performances**. We measure the DCI-Disentanglement metric on the 10-dimensional representation for $\beta$-VAE, PCL, SlowVAE and Ada-GVAE and the corresponding $R^2$-score on the downstream performance. All p-values are below 0.01 except for composition on Shapes3D which has p-value=0.14. Note that, we here provide Spearman's rank correlation instead Pearson as the p-values are slightly lower.

| Data set | Modification | R-squared Test | Modification | R-squared Test |
|---|---|---|---|---|
| dSprites | random | 1.000 | random + sign-flip | 1.000 |
| dSprites | composition | 1.000 | composition + sign-flip | 1.000 |
| dSprites | interpolation | 1.000 | interpolation + sign-flip | 1.000 |
| dSprites | extrapolation | 1.000 | extrapolation + sign-flip | 0.999 |
| Shapes3D | random | 1.000 | random + sign-flip | 1.000 |
| Shapes3D | composition | 1.000 | composition + sign-flip | 1.000 |
| Shapes3D | interpolation | 1.000 | interpolation + sign-flip | 1.000 |
| Shapes3D | extrapolation | 1.000 | extrapolation + sign-flip | 1.000 |
| MPI3D-Real | random | 1.000 | random + sign-flip | 1.000 |
| MPI3D-Real | composition | 1.000 | composition + sign-flip | 0.996 |
| MPI3D-Real | interpolation | 1.000 | interpolation + sign-flip | 1.000 |
| MPI3D-Real | extrapolation | 0.999 | extrapolation + sign-flip | 0.997 |

Table 1: **Performances of the readout-MLP on the ground-truth.**

## A CONNECTION BETWEEN READOUT PERFORMANCE AND DISENTANGLEMENT OF THE REPRESENTATION

Here, we narrow down the root cause of the limited extrapolation performance of disentanglement models in the OOD settings as observed in Figs. 4 and 11. More precisely, we investigate how the readout-MLP would perform on a perfectly disentangled representation. Therefore, we train our readout MLP directly on the ground-truth factors of variation for all possible test-train splits described in Fig. 2 and measured the $R^2$-score test error for each split. Here, the MLP only has to learn the identity function. In a slightly more evolved setting, termed *sign-flip*, we switched the sign input to train the readout-MLP on a mapping from -ground-truth to ground-truth. This mimics the identifiability guarantees of models like SlowVAE which are up to permutation and sign flips under certain assumptions. The R-squared for all settings in Table 1 are $> .99$, therefore the readout model should not be the limitation for OOD generalization in our setting if the representation is identified up to permutation and sign flips. Note that this experiment does not cover disentanglement up to point-wise nonlinearities or linear/ affine transformations as required by other models.

## B CELEBGLOW DATASET

The current disentanglement datasets such as dSprites, Shapes3D, MPI3D, and others are constructed based on highly controlled environments (Matthey et al., 2017; Kim & Mnih, 2018; Gondal et al., 2019). Here, common factors of variations are rotations or scaling of simple geometric objects, such as a square. For a more intuitive investigation of other factors, we created the CelebGlow dataset. Here, the factors of variations are smiling, blondness and age. Samples are shown in Fig. 8. Note that we rely on the Glow model instead of taking a real-world dataset, as this allows for a gradual control of individual factors of variation.

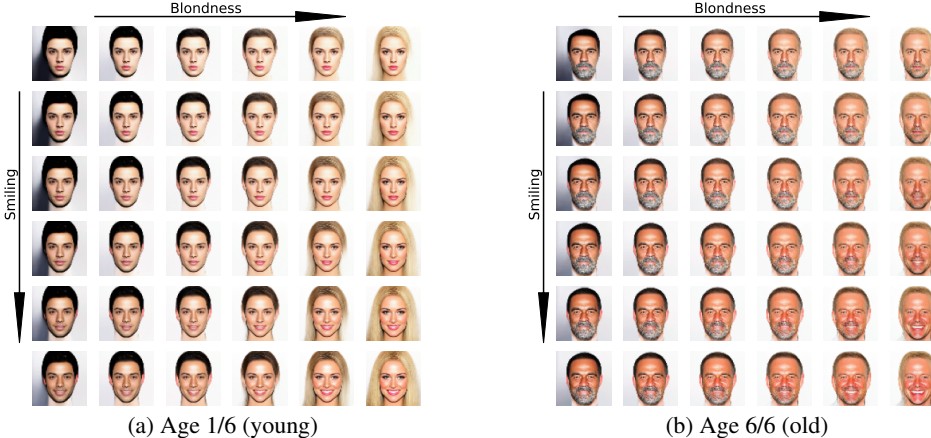

(a) Age 1/6 (young)                    (b) Age 6/6 (old)

Figure 8: **CelebGlow Dataset**

The CelebGlow dataset is created based on the invertible generative neural network of Kingma et al. (Kingma & Dhariwal, 2018). We used their provided network[2] that is pretrained on the Celeb-HQ dataset, and has labelled directions in the model-latent space that correspond to specific attributes of the dataset. Based on this latent space, we created the dataset as follows:

1. In the latent space of the model, we sample from a high dimensional Gaussian with zero mean and a low standard deviation of 0.3 to avoid too much variability.

2. Next, we perform a latent walk into the directions that correspond to "Smiling", "Age" and "Blondness" in image space. To estimate the spacing, we rely on the function `manipulate_range`[3]. We perform 6 steps along each axis and all combinations (6x6x6 cube). As a scale parameter to the function, we use 0.8. Those factors were chosen s.t. the images differ significantly, but also to stay in the valid range of the model based on visual inspection.

3. We pass all latent coordinates through the glow network in the generative direction.

4. We further down-sample the images from 256x256x3 to 64x64x3 to match the resolution of common disentanglement datasets.

5. Finally, we store each image and the corresponding factor combination.

This procedure is repeated for 1000 samples to get $6 * 6 * 6 * 1000 = 216000$ samples in total, which is around the same size as other common datasets.

## C   HYPERPARAMETER TUNING ABLATION

As described in the implementation details, we use common values from the literature to train the proposed models. Here, we investigate effects of such hyperparameters on the CNN architecture. Due to the combinatorial complexity, we do not perform a search for other architectures. As hyperparameters, we varied the number or training iterations (3 different numbers of iterations), we introduced 5 different strengths of regularization, 2 different depths for the CNN architecture [6 layers, 9 layers] and ran multiple random seeds for each combination.

The results on the extrapolation test on MPI3D set are shown in Fig. 9. Given this hyperparameter search, we find no improvement over our reported numbers for the CNN.

---

[2]The network can be found at: https://github.com/openai/glow/blob/master/demo/script.sh#L24

[3]https://github.com/openai/glow/blob/master/demo/model.py#L219

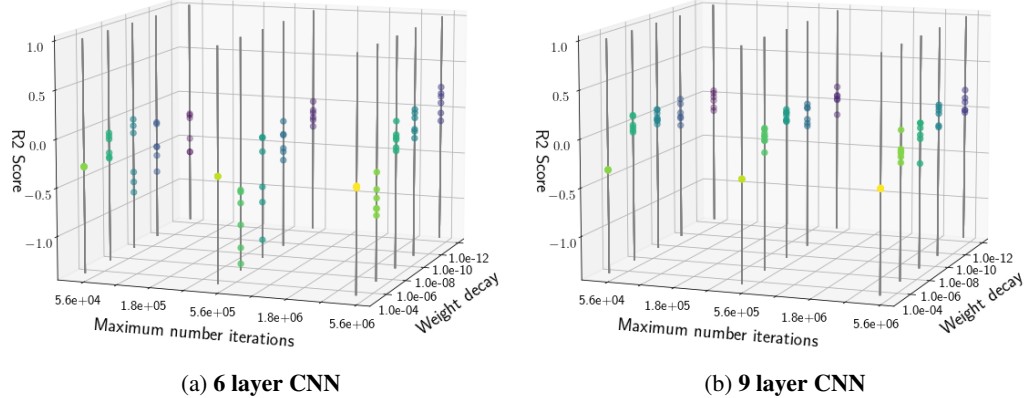

(a) **6 layer CNN**              (b) **9 layer CNN**

Figure 9: **Hyperparameter search on MPI3D for a CNN**.

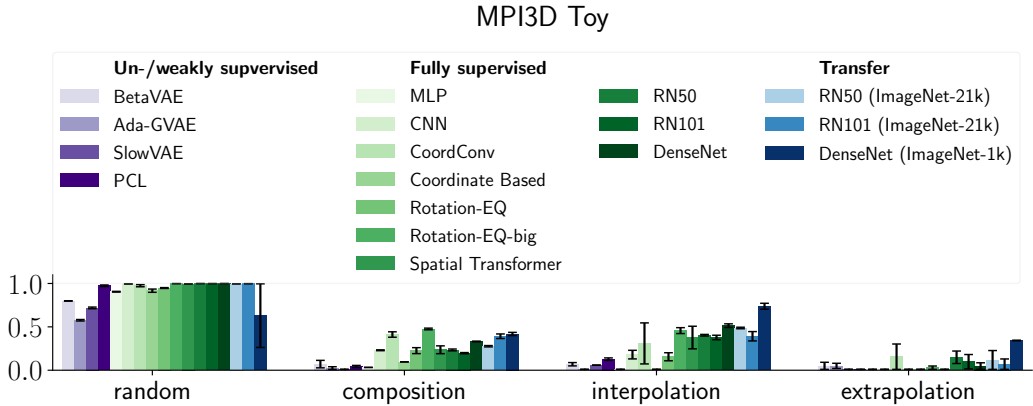

Figure 10: $R^2$**-score on MPI3D-Synthetic**.

## D    REAL VERSUS SYNTHETIC DATASET

To narrow down the question "why the generalization capabilities drop on real-world dataset MPI3D?", we run a comparison on MPI3D dataset with real and synthetic images.

The results on the MPI3D dataset with synthetic images is shown in Fig. 10 and Table 10. Comparing this with R-squared performances to MPI3D with real-world images (Fig. 4 and Table 9), we observe that the results do not change significantly (most results are in a 1-2sigma range). We conclude that the larger drops in performance on MPI3D compared to Shapes3D or dSprites, are not due to the real images as opposed to synthetic images. Instead, we hypothesize that it is due to the more realistic setup of the MPI3D dataset itself. For instance, it contains complex factors like rotation in 3D projected on 2D. Here, occluded parts of objects have to be guessed based on certain symmetry assumptions.

## E    ABLATION ON NON-AMBIGUOUS DSPRITES

The setup of dSprites is non-injective, as different rotations map to the same image. E.g., the square at a rotation $90°$ is identical to the one rotated by $180°$ and therefore ambiguous. Thus, the training process is noisy. In an ablation study, we controlled for this by constraining the rotations to lie in $[0, 90)$. We again ran all our proposed models and report the $R^2$-Score in Fig. 11b.

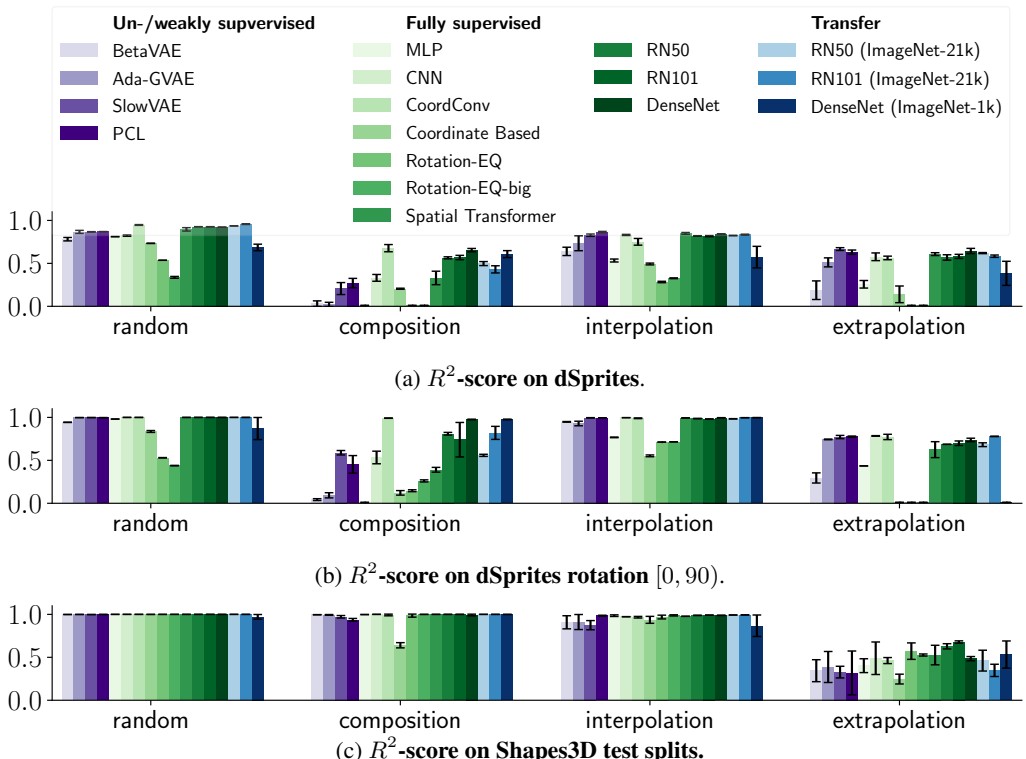

(a) $R^2$-**score on dSprites**.

(b) $R^2$-**score on dSprites rotation** $[0, 90]$.

(c) $R^2$-**score on Shapes3D test splits**.

Figure 11: $R^2$-**score on various splits.**

Comparing the new results with the original dSprites results shows: First, for the random test-train split, resolving the rotational ambiguity leads to almost perfect performance (close to $100\%$ R-squared scores for most models). In the previous dSprites setup with rotational ambiguity, top accuracies are around 70-95% R-squared scores for most models. Second, large drops in performance can still be observed when we move towards the systematic out-of-distribution splits (composition, interpolation, and extrapolation). Also, our insights on how models extrapolate remain the same. Lastly, for the random split, the Rotation-EQ model shows non-perfect performance. Tracing this error to individual factors, it turns out this is due to limited capabilities in predicting the x, y positions. We hypothesize that this is due to limitations of convolutions in propagating spatial positions, as discussed in (Liu et al., 2018). The DenseNet performs perfectly on the train set and might be overfitting.

We conclude that the rotational ambiguity explains the drops on the random split. However, the clear drops in performance on the systematic splits remain nonetheless. Thus, the analysis we perform in the paper and the conclusions we draw remain the same.

## F   DATA AUGMENTATIONS

We investigate the effects of data augmentation during training time on the generalization performance in the extrapolation setting of our proposed benchmark.

As data augmentations, we applied random erasing, Gaussian Noise, small shearings, and blurring. Note that we could not use arbitrary augmentations. For instance, shift augmentations would lead to ambiguities with the "shift" factor in dSprites. Next, we trained CNNs with and without data augmentations on all four datasets (dSprites, Shapes3D, MPI3D, CelebGlow) on the extrapolation splits with multiple random seeds.

The results are visualized in Fig. 12. For the mean performance, we observe no significant improvement by adding augmentations. However, the overall spread of the scores seems to decrease given augmentations on some datasets. We explain this by the fact that the augmentations enforce cer-

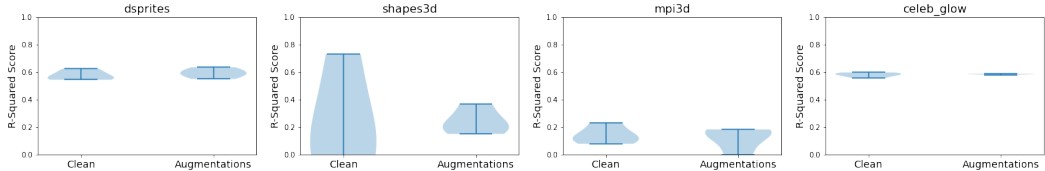

Figure 12: $R^2$**-score on data augmenations**. We depict the performance on the extrapolation setting with and without data augmentations for a CNN network with various random seeds on our considered datasets.

tain invariances, narrowing the solution space of optimal training solutions by providing a further specification (specification in the sense of D'Amour et al. (2020)).

## G    PERFORMANCE WITH RESPECT TO INDIVIDUAL FACTORS

We here try to attribute the performance losses to individual OOD factors (see §5.2). Thus, on the extrapolation setting, we modify the test-splits such that only a single factor is out-of-distribution. Next, we measure the overall performance across models (all fully supervised and transfer models) to demonstrate the effect of this factor. The results are depicted for all models in Fig. 13. Overall, factors like "height" on MPI3D that control the viewing of the camera and, subsequently, change attributes like the absolute position in the image of other factors (e.g., the tip of the robot arm) have a high effect.

## H    IMPLEMENTATION DETAILS

### H.1    DATA SETS

Each dataset consists of multiple factors of variation and every possible combination of factors generates a corresponding image. Here, we list all datasets and their corresponding factor ranges. Note, to estimate the reported $R^2$-score, we normalize the factors by dividing each factor $\mathbf{y}_i$ by $|\mathbf{y}_i^{\max} - \mathbf{y}_i^{\min}|$, i.e., all factors are in the range $[0, 1]$. *dSprites* (Matthey et al., 2017), represents some low resolution binary images of basic shapes with the 5 FoVs shape $\{0, 1, 2\}$, scale $\{0, ..., 4\}$, orientation[4] $\{0, ..., 39\}$, x-position $\{0, ..., 31\}$, and y-position $\{0, ..., 31\}$. Next, *Shapes3D* (Kim & Mnih, 2018) which is a similarly popular dataset with 3D shapes in a room scenes defined by the 6 FoVs floor color $\{0, ..., 9\}$, wall color $\{0, ..., 9\}$, object color $\{0, ..., 9\}$, object size $\{0, ..., 7\}$, object type $\{0, ..., 3\}$ and azimuth $\{0, ..., 14\}$. Lastly, we consider the challenging and more realistic dataset *MPI3D* (Gondal et al., 2019) containing real images of physical 3D objects attached to a robotic finger generated by 7 FoVs color $\{0, ..., 5\}$, shape $\{0, ..., 5\}$, size $\{0, 1\}$, height $\{0, 1, 2\}$, background color $\{0, 1, 2\}$, x-axis $\{0, ..., 39\}$ and y-axis $\{0, ..., 39\}$.

### H.2    DATA SET SPLITS

Each dataset is complete in the sense that it contains all possible combinations of factors of variation. Thus, the interpolation and extrapolation test-train splits are fully defined by specifying which factors are exclusively in the test set. Starting from all possible combinations, if a given factor value

---

[4]Note that this dataset contains a non-injective generative model as square and ellipses have multiple rotational symmetries.

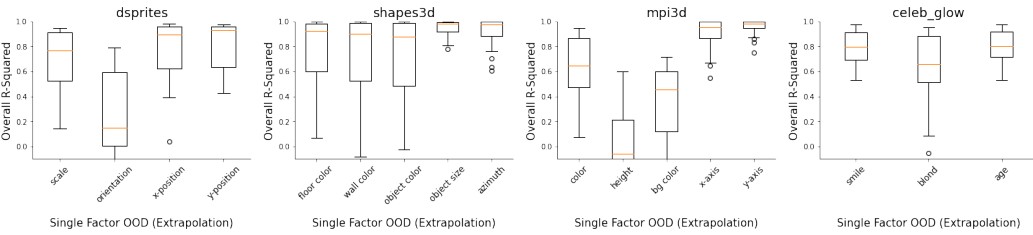

Figure 13: $R^2$**-score on in individual factors**. Extrapolation performance across models when only a single factor (x-axis) is OOD.

is defined to be exclusively in the test set, the corresponding image is part of the test set. E.g. for the extrapolation case in dSprites, all images containing x-positions > 24 are part of the test set and the train set its respective complement $D \backslash D_{test}$. Composition can be defined equivalently to extrapolation but with interchanged test and train sets. The details of the splits are provided in table Tables 2 and 3. The resulting train vs. test sample number ratios are roughly $30 : 70$. See Table 4. We will release the test and train splits to allow for a fair comparison and benchmarking for future work.

For the setting where only a single factor is OOD, we formally define this as

$$\mathcal{D}_{one\text{-}ood} = \{(\mathbf{y}^k, \mathbf{x}^k) \in \mathcal{D}_{\text{te}} \mid \exists! i \in \mathbb{N} \; s.t. \; y_i^k \neq y_i^l \; \forall (\mathbf{y}^l, \mathbf{x}^l) \in \mathcal{D}_{\text{tr}}\}. \tag{3}$$

Here, we used the superscript indices to refer to a sample and the subscript to denote the factor. Note that the defined set is only nonempty in the interpolation and extrapolation settings.

## H.3 TRAINING

All models are implemented using PyTorch 1.7. If not specified otherwise, the hyperparameters correspond to the default library values.

**Un-/ weakly supervised** For the un-/weakly supervised models, we consider 10 random seeds per hyperparameter setup. As hyperparameters, we optimize one parameter of the learning objective per model similar to Table 2 from Locatello et al. (Locatello et al., 2020a). For the SlowVAE, we took the optimal values from Klindt et al. (Klindt et al., 2020) and tuned for $\gamma \in \{1, 5, 10, 15, 20, 25\}$. The PCL model itself does not have any hyperparameters (Hyvarinen & Morioka, 2017). For simplicity, we determine the optimal setup in a supervised manner by measuring the DCI-Disentanglement score (Eastwood & Williams, 2018) on the training split. The PCL and SlowVAE models are trained on pairs of images that only differ sparsely in their underlying factors of variation following a Laplace transition distribution, the details correspond to the implementation [5] of Klindt et al. (Klindt et al., 2020). The Ada-GVAE models are trained on pairs of images that differ uniformly in a single, randomly selected factor. Other factors are kept fixed. This matches the strongest model from Locatello et al. (Locatello et al., 2020a) implemented on GitHub[6]. All $\beta$-VAE models are trained in an unsupervised manner. All un- and weakly supervised models are trained with the Adam optimizer with a learning rate of $0.0001$. We train each model for $500,000$ iterations with a batch size of 64, which for the weakly supervised models, corresponds to 64 pairs. Lastly, we train a supervised readout model on top of the latents for 8 epochs with the Adam optimizer on the full corresponding training dataset and observe convergence on the training and test datasets - no overfitting was observed.

**Fully supervised:** All fully supervised models are trained with the same training scheme. We use the Adam optimizer with a learning rate of $0.0005$. The only exception is DenseNet, which is trained with a learning rate of $0.0001$, as we observe divergences on the training loss with the higher learning rate. We train each model with three random seeds for $500,000$ iterations with a batch size of $b = 64$. As a loss function, we consider the mean squared error $\text{MSE} = \sum_{j=0}^{b} ||\mathbf{y}_j - f_j(\mathbf{x})||_2^2 / b$ per mini-batch.

**Transfer learning:** The pre-trained models are fine-tuned with the same loss as the fully supervised models. We train for $50,000$ iterations and with a lower learning rate of $0.0001$. We fine-tune all model weights. As an ablation, we also tried only training the last layer while freezing the other weights. In this setting, we consistently observed worse results and, therefore, do not include them in this paper.

## H.4 MODEL IMPLEMENTATIONS

Here, we shortly describe the implementation details required to reproduce our model implementation. We denote code from Python libraries in grey. If not specified otherwise, the default parameters and nomenclature correspond to the PyTorch 1.7 library.

---

[5]https://github.com/bethgelab/slow_disentanglement/blob/master/scripts/dataset.py#L94

[6]https://github.com/google-research/disentanglement_lib/blob/master/disentanglement_lib/methods/weak/weak_vae.py#L62 and https://github.com/google-research/disentanglement_lib/blob/master/disentanglement_lib/methods/weak/weak_vae.py#L317

| | dataset | split | name | exclusive test factors |
|---|---|---|---|---|
| 0 | dSprites | interpolation | shape | {} |
| 1 | dSprites | interpolation | scale | {1, 4} |
| 2 | dSprites | interpolation | orientation | {32, 2, 37, 7, 12, 17, 22, 27} |
| 3 | dSprites | interpolation | x-position | {2, 7, 11, 15, 20, 24, 29} |
| 4 | dSprites | interpolation | y-position | {2, 7, 11, 15, 20, 24, 29} |
| 5 | dSprites | extrapolation | shape | {} |
| 6 | dSprites | extrapolation | scale | {4, 5} |
| 7 | dSprites | extrapolation | orientation | {32, 33, 34, 35, 36, 37, 38, 39} |
| 8 | dSprites | extrapolation | x-position | {25, 26, 27, 28, 29, 30, 31} |
| 9 | dSprites | extrapolation | y-position | {25, 26, 27, 28, 29, 30, 31} |
| 10 | Shapes3D | interpolation | floor color | {2, 7} |
| 11 | Shapes3D | interpolation | wall color | {2, 7} |
| 12 | Shapes3D | interpolation | object color | {2, 7} |
| 13 | Shapes3D | interpolation | object size | {2, 5} |
| 14 | Shapes3D | interpolation | object type | {} |
| 15 | Shapes3D | interpolation | azimuth | {2, 12, 7} |
| 16 | Shapes3D | extrapolation | floor color | {8, 9} |
| 17 | Shapes3D | extrapolation | wall color | {8, 9} |
| 18 | Shapes3D | extrapolation | object color | {8, 9} |
| 19 | Shapes3D | extrapolation | object size | {6, 7} |
| 20 | Shapes3D | extrapolation | object type | {} |
| 21 | Shapes3D | extrapolation | azimuth | {12, 13, 14} |
| 22 | MPI3D | interpolation | color | {3} |
| 23 | MPI3D | interpolation | shape | {} |
| 24 | MPI3D | interpolation | size | {} |
| 25 | MPI3D | interpolation | height | {1} |
| 26 | MPI3D | interpolation | background color | {1} |
| 27 | MPI3D | interpolation | x-axis | {24, 34, 5, 15} |
| 28 | MPI3D | interpolation | y-axis | {24, 34, 5, 15} |
| 29 | MPI3D | extrapolation | color | {5} |
| 30 | MPI3D | extrapolation | shape | {} |
| 31 | MPI3D | extrapolation | size | {} |
| 32 | MPI3D | extrapolation | height | {2} |
| 33 | MPI3D | extrapolation | background color | {2} |
| 34 | MPI3D | extrapolation | x-axis | {36, 37, 38, 39} |
| 35 | MPI3D | extrapolation | y-axis | {36, 37, 38, 39} |
| 36 | CelebGlow | interpolation | person | {} |
| 37 | CelebGlow | interpolation | smile | {1, 4} |
| 38 | CelebGlow | interpolation | blond | {1, 4} |
| 39 | CelebGlow | interpolation | age | {1, 4} |
| 40 | CelebGlow | extrapolation | person | {} |
| 41 | CelebGlow | extrapolation | smile | {4, 5} |
| 42 | CelebGlow | extrapolation | blond | {4, 5} |
| 43 | CelebGlow | extrapolation | age | {4, 5} |

Table 2: **Interpolation and extrapolation splits.**

|    | dataset  | split       | name             | exclusive train factors |
|----|----------|-------------|------------------|-------------------------|
| 0  | dSprites | composition | shape            | {}                      |
| 1  | dSprites | composition | scale            | {}                      |
| 2  | dSprites | composition | orientation      | {0, 1, 2, 3}            |
| 3  | dSprites | composition | x-position       | {0, 1, 2}               |
| 4  | dSprites | composition | y-position       | {0, 1, 2}               |
| 5  | Shapes3D | composition | floor color      | {0}                     |
| 6  | Shapes3D | composition | wall color       | {0}                     |
| 7  | Shapes3D | composition | object color     | {0}                     |
| 8  | Shapes3D | composition | object size      | {}                      |
| 9  | Shapes3D | composition | object type      | {}                      |
| 10 | Shapes3D | composition | azimuth          | {0}                     |
| 11 | MPI3D    | composition | color            | {}                      |
| 12 | MPI3D    | composition | shape            | {}                      |
| 13 | MPI3D    | composition | size             | {}                      |
| 14 | MPI3D    | composition | height           | {}                      |
| 15 | MPI3D    | composition | background color | {}                      |
| 16 | MPI3D    | composition | x-axis           | {0, 1, 2, 3, 4, 5}      |
| 17 | MPI3D    | composition | y-axis           | {0, 1, 2, 3, 4, 5}      |

Table 3: **Composition splits.**

|    | dataset  | split         | % test | % train | Total samples |
|----|----------|---------------|--------|---------|---------------|
| 0  | dSprites | random        | 32.6   | 67.4    | 737280        |
| 1  | dSprites | composition   | 26.1   | 73.9    | 737280        |
| 2  | dSprites | interpolation | 32.6   | 67.4    | 737280        |
| 3  | dSprites | extrapolation | 32.6   | 67.4    | 737280        |
| 4  | Shapes3D | random        | 30.7   | 69.3    | 480000        |
| 5  | Shapes3D | composition   | 32.0   | 68.0    | 480000        |
| 6  | Shapes3D | interpolation | 30.7   | 69.3    | 480000        |
| 7  | Shapes3D | extrapolation | 30.7   | 69.3    | 480000        |
| 8  | MPI3D    | random        | 30.0   | 70.0    | 1036800       |
| 9  | MPI3D    | composition   | 27.8   | 72.2    | 1036800       |
| 10 | MPI3D    | interpolation | 30.0   | 70.0    | 1036800       |
| 11 | MPI3D    | extrapolation | 30.0   | 70.0    | 1036800       |

Table 4: **Test train ratio.**

The un- and weakly supervised models $\beta$-**VAE**, **Ada-GVAE** and **SlowVAE** all use the same encoder-decoder architecture as Locatello et al. (Locatello et al., 2020a). The **PCL** model uses the same architecture as the encoder as well and with the same readout structure for the contrastive loss as used by Hyvärinen et al. (Hyvarinen & Morioka, 2017). For the supervised readout MLP, we use the sequential model `[Linear(10, 40), ReLU(), Linear(40, 40), ReLU(40, 40), Linear(40, 40), ReLU(), Linear(40, number-factors)]`.

The **MLP** model consists of `[Linear(64*64*number-channels, 90), ReLU(), Linear(90, 90), ReLU(), Linear(90, 90), ReLU(), Linear(90, 90), ReLU(), Linear(90, 45), ReLU(), Linear(22, number-factors)]`. The architecture is chosen such that it has roughly the same number of parameters and layers as the CNN.

The **CNN** architecture corresponds the one used by Locatello et al. (Locatello et al., 2020a). We only adjust the number of outputs to match the corresponding datasets.

The **CoordConv** consists of a *CoordConv2D* layer following the PyTorch implementation[7] with 16 output channels. It is followed by 5 ReLU-Conv layers with 16 in- and output channels each and a MaxPool2D layer. The final readout consists of `[Linear(32, 32), ReLU(), Linear(32, number-factors)]`.

The **SetEncoder** concatenates each input pixel with its $i, j$ pixel coordinates normalized to $[0, 1]$. All concatenated pixels (i, j, pixel-value) are subsequently processed with the same network which consists of `[Linear(2+number-channels), ReLU(), Linear(40, 40), ReLU(), Linear(40, 20), ReLU()]`. This is followed by a mean pooling operation per image which guarantees an invariance over the order of the inputs, i.e. one could shuffle all inputs and the output would remain the same. As a readout, it follows a sequential fully connected network consisting of `[Linear(20, 20), ReLU(), Linear(20, 20), ReLU(), Linear(20, number-factors)]`.

The rotationally equivariant network **RotEQ** is similar to the architecture from Locatello et al. (Locatello et al., 2020a). One difference is that it uses the R2Conv module[8] from Weiler et al. (Weiler & Cesa, 2019) instead of the PyTorch *Conv2d* with an 8-fold rotational symmetry. We thus decrease the number of feature maps by a factor of 8, which roughly corresponds to the same computational complexity as the CNN. We provide a second version which does not decrease the number of feature maps and, thus, has the same number of trainable parameters as the CNN but a higher computational complexity. We refer to this version as **RotEQ-big.**

To implement the spatial transformer (**STN**) (Wu et al., 2019), we follow the PyTorch tutorial implementation[9] which consists of two steps. In the first step, we estimate the parameters of a (2, 3)-shaped affine matrix using a sequential neural network with the following architecture `[Conv2d(number_channels, 8, kernel_size=7), MaxPool2d(2, stride=2), ReLU(), Conv2d(8, 10, kernel_size=5), MaxPool2d(2, stride=2), ReLU(), Conv2d(10, 10, kernel_size=6), MaxPool2d(2, stride=2), ReLU(), Linear(10*3*3, 31), ReLU(), Linear(32, 3*2)]`. In the second step, the input image is transformed by the estimated affine matrix and subsequently processed by a CNN which has the same architecture as the CNN described above.

For the transfer learning models ResNet50 (**RN50**) and ResNet101 (**RN101**) pretrained on ImageNet-21k (IN-21k), we use the big-transfer (Kolesnikov et al., 2020) implementation[10]. For the RN50, we download the weights with the tag `"BiT-M-R50x1"`, and for the RN101, we use the tag `"BiT-M-R101x3"`. For the **DenseNet** trained on ImageNet-1k (IN-1k), we used the weights from `densenet121`. For all transfer learning methods, we replace the last layer of the pre-trained models with a randomly initialized linear layer which matches the number of outputs to the number

---

[7]https://github.com/walsvid/CoordConv
[8]https://github.com/QUVA-Lab/e2cnn
[9]https://pytorch.org/tutorials/intermediate/spatial_transformer_tutorial.html
[10]https://colab.research.google.com/github/google-research/big_transfer/blob/master/colabs/big_transfer_pytorch.ipynb and for the weights https://storage.googleapis.com/bit_models/{bit_variant}.npz

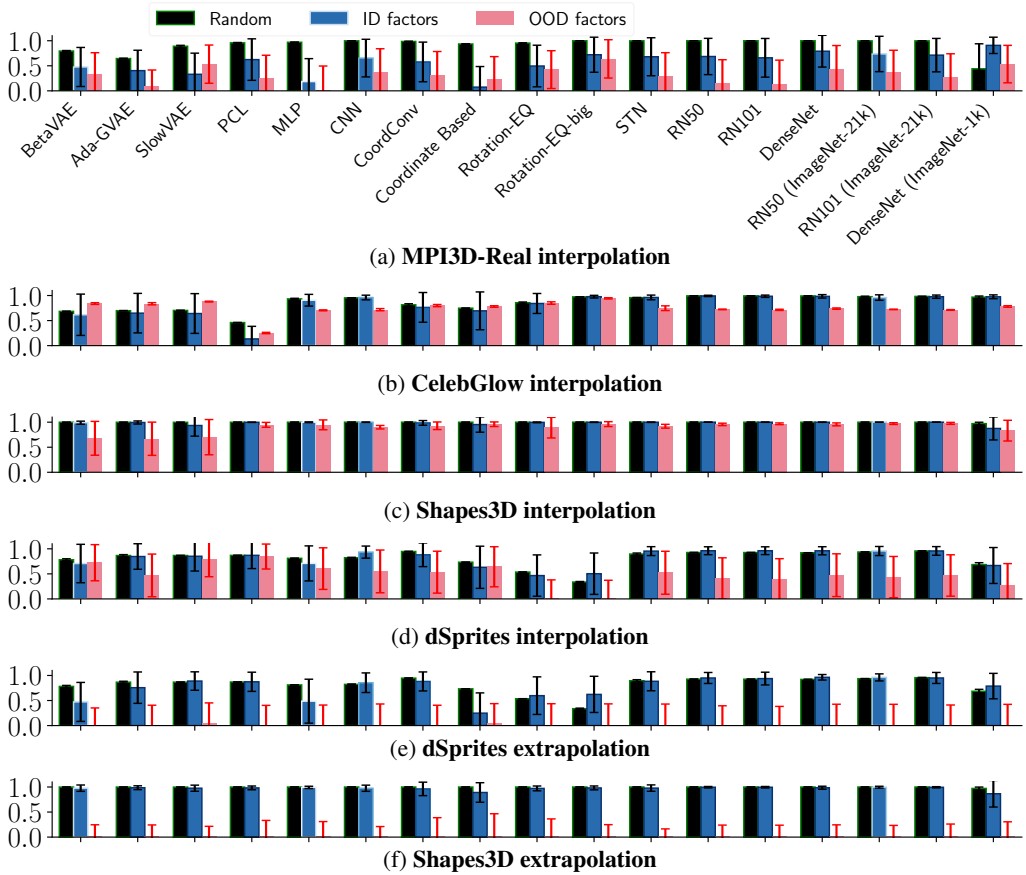

Figure 14: **Interpolation / Extrapolation and modularity.**

of factors in each dataset. As an ablation, we also provide a randomly initialized version for each transfer learning model.

### H.5 COMPUTE

All models are run on the NVIDIA T4 Tensor Core GPUs on the AWS g4dn.4xlarge instances with an approximate total compute of 20 000 GPUh. To save computational cost, we gradually increased the number of seeds until we achieved acceptable p-values of $\leq 0.05$. In the end, we have 3 random seeds per supervised model and 10 random seeds per hyperparameter setting for the un and weakly supervised models.

### I ADDITIONAL RESULTS

| modification models | random | composition | interpolation | extrapolation |
|---|---|---|---|---|
| BetaVAE | 78.2± 2.1 | 0.1± 6.5 | 64.0± 5.1 | 18.3± 12.3 |
| Ada-GVAE | 86.6± 1.9 | -1.4± 5.5 | 73.4± 9.2 | 51.2± 5.5 |
| SlowVAE | 86.7± 0.2 | 20.7± 7.3 | 82.8± 1.4 | 66.7± 1.7 |
| PCL | 87.0± 0.1 | 27.1± 5.5 | 86.4± 0.9 | 63.0± 2.5 |
| MLP | 81.1± 0.2 | -10.6± 1.3 | 53.3± 2.1 | 25.7± 5.4 |
| CNN | 82.3± 0.9 | 33.1± 4.8 | 83.1± 0.8 | 57.7± 5.5 |
| CoordConv | 94.7± 0.6 | 67.7± 5.1 | 75.1± 4.8 | 56.3± 2.6 |
| Coordinate Based | 73.3± 0.3 | 20.4± 0.8 | 49.3± 1.3 | 8.8± 20.5 |
| Rotation-EQ | 53.6± 0.1 | -12.9± 7.1 | 28.3± 1.0 | -23.8± 4.1 |
| Rotation-EQ-big | 33.7± 1.3 | -8.6± 1.6 | 32.7± 0.5 | -25.9± 1.2 |
| Spatial Transformer | 89.6± 2.4 | 33.0± 9.6 | 84.9± 1.3 | 60.8± 2.0 |
| RN50 | 92.7± 0.1 | 56.5± 1.4 | 82.1± 0.1 | 56.9± 3.9 |
| RN101 | 92.6± 0.1 | 56.8± 3.1 | 81.7± 0.8 | 58.1± 2.9 |
| DenseNet | 92.2± 0.2 | 65.4± 2.4 | 84.3± 0.2 | 64.4± 3.7 |
| RN50 (ImageNet-21k) | 93.6± 0.2 | 49.7± 2.9 | 82.5± 0.3 | 62.0± 0.8 |
| RN101 (ImageNet-21k) | 95.7± 0.5 | 43.0± 4.8 | 83.5± 0.6 | 58.3± 1.6 |
| DenseNet (ImageNet-1k) | 68.5± 5.4 | 60.8± 5.7 | 57.3± 17.7 | 38.4± 19.8 |

Table 5: $R^2$-score on dSprites

| modification models | random | composition | interpolation | extrapolation |
|---|---|---|---|---|
| BetaVAE | 94.3+- 0.3 | 4.3+- 1.5 | 94.8+- 0.5 | 29.5+- 8.4 |
| Ada-GVAE | 99.8+- 0.0 | 9.3+- 4.3 | 92.9+- 3.8 | 74.4+- 0.6 |
| SlowVAE | 99.8+- 0.0 | 58.9+- 3.7 | 99.5+- 0.3 | 77.1+- 2.7 |
| PCL | 99.7+- 0.0 | 45.3+- 14.4 | 99.6+- 0.0 | 77.5+- 1.1 |
| MLP | 98.1+- 0.1 | -8.8+- 0.2 | 76.7+- 0.5 | 43.5+- 0.1 |
| CNN | 100.0+- 0.0 | 53.3+- 10.4 | 99.7+- 0.0 | 78.4+- 0.5 |
| CoordConv | 100.0+- 0.0 | 99.1+- 0.3 | 99.0+- 0.6 | 77.2+- 4.4 |
| Coordinate Based | 83.6+- 1.5 | 12.1+- 3.7 | 55.2+- 1.5 | -11.5+- 15.6 |
| Rotation-EQ | 53.0+- 0.3 | 14.6+- 1.6 | 71.3+- nan | -6.9+- 0.8 |
| Rotation-EQ-big | 43.9+- 0.4 | 26.0+- 1.7 | 71.5+- 0.0 | -1.9+- 1.3 |
| Spatial Transformer | 100.0+- 0.0 | 39.0+- 4.0 | 99.4+- 0.2 | 62.5+- 13.1 |
| RN50 | 100.0+- 0.0 | 81.0+- 2.2 | 98.7+- 0.4 | 68.6+- 0.1 |
| RN101 | 100.0+- 0.0 | 74.0+- 28.4 | 98.0+- 0.4 | 69.9+- 3.7 |
| DenseNet | 100.0+- 0.0 | 97.6+- 0.5 | 99.6+- 0.2 | 73.8+- 2.8 |
| RN50 (ImageNet-21k) | 100.0+- 0.0 | 55.8+- 1.8 | 98.3+- 0.3 | 68.2+- 3.1 |
| RN101 (ImageNet-21k) | 100.0+- 0.0 | 82.0+- 10.7 | 99.6+- 0.1 | 77.9+- 0.4 |
| DenseNet (ImageNet-1k) | 87.0+- 18.2 | 97.6+- 0.5 | 99.8+- 0.1 | -420.5+- 504.1 |

Table 6: $R^2$-score on dSprites rotation $[0, 90)$

| modification models | random | composition | interpolation | extrapolation |
|---|---|---|---|---|
| BetaVAE | 99.9± 0.1 | 99.6± 0.2 | 90.8± 8.0 | 34.4± 13.5 |
| Ada-GVAE | 99.9± 0.0 | 99.4± 0.4 | 91.1± 9.2 | 37.6± 21.6 |
| SlowVAE | 99.8± 0.1 | 97.2± 1.5 | 87.4± 5.7 | 32.8± 7.2 |
| PCL | 99.9± 0.0 | 93.6± 1.6 | 98.5± 0.5 | 29.8± 29.1 |
| MLP | 100.0± 0.0 | 99.8± 0.2 | 98.5± 1.2 | 40.3± 9.9 |
| CNN | 100.0± 0.0 | 100.0± 0.0 | 97.3± 0.1 | 48.9± 23.2 |
| CoordConv | 100.0± 0.0 | 99.3± 1.2 | 96.7± 1.1 | 46.2± 4.3 |
| Coordinate Based | 100.0± 0.0 | 64.0± 3.7 | 93.6± 5.0 | 24.7± 7.0 |
| Rotation-EQ | 100.0± 0.0 | 98.5± 2.2 | 96.9± 2.7 | 57.2± 11.7 |
| Rotation-EQ-big | 100.0± 0.0 | 100.0± 0.0 | 98.8± 0.8 | 52.7± 1.5 |
| Spatial Transformer | 100.0± 0.0 | 100.0± 0.0 | 97.8± 0.1 | 52.7± 13.9 |
| RN50 | 100.0± 0.0 | 100.0± 0.0 | 98.8± 0.3 | 62.8± 3.7 |
| RN101 | 100.0± 0.0 | 100.0± 0.0 | 99.1± 0.1 | 67.8± 1.7 |
| DenseNet | 100.0± 0.0 | 99.3± 1.2 | 98.9± 0.3 | 48.5± 3.0 |
| RN50 (ImageNet-21k) | 100.0± 0.0 | 100.0± 0.0 | 99.3± 0.1 | 46.1± 14.8 |
| RN101 (ImageNet-21k) | 100.0± 0.0 | 100.0± 0.0 | 99.4± 0.2 | 34.8± 8.3 |
| DenseNet (ImageNet-1k) | 97.1± 3.8 | 100.0± 0.0 | 86.8± 17.7 | 53.2± 22.3 |

Table 7: $R^2$-score on Shapes3D

| modification models | random | composition | interpolation | extrapolation |
|---|---|---|---|---|
| BetaVAE | 68.4+- 0.6 | 43.7+- 3.1 | 66.2+- 0.1 | 43.5+- 0.3 |
| Ada-GVAE | 69.8+- 0.4 | 48.6+- 2.7 | 68.4+- 0.6 | 44.5+- 0.5 |
| SlowVAE | 70.3+- 0.6 | 53.9+- 2.7 | 69.5+- 0.0 | 43.7+- 1.0 |
| PCL | 46.0+- 0.1 | -7.0+- 1.4 | 19.7+- 0.1 | -48.4+- 2.6 |
| MLP | 93.3+- 1.7 | 75.3+- 2.2 | 82.0+- 0.6 | 50.0+- 2.7 |
| CNN | 95.4+- 0.0 | 75.9+- 1.7 | 86.9+- 1.3 | 57.9+- 3.0 |
| CoordConv | 81.5+- 3.0 | 57.7+- 1.8 | 74.4+- 2.4 | 33.5+- 6.7 |
| Coordinate Based | 74.8+- 0.3 | 58.0+- 3.9 | 68.8+- 0.5 | 22.7+- 1.8 |
| Rotation-EQ | 85.7+- 1.9 | 59.9+- 2.9 | 82.0+- 2.1 | 44.1+- 5.1 |
| Rotation-EQ-big | 97.3+- 0.0 | 81.7+- 1.6 | 96.1+- 0.1 | 66.4+- 0.3 |
| Spatial Transformer | 95.8+- 0.1 | 76.8+- 0.1 | 88.0+- 0.9 | 57.7+- 0.5 |
| RN50 | 99.3+- 0.3 | 79.5+- 2.8 | 89.3+- 0.2 | 48.6+- 1.0 |
| RN101 | 99.3+- 0.1 | 81.5+- 2.0 | 88.6+- 0.1 | 49.1+- 0.7 |
| DenseNet | 99.2+- 0.3 | 83.9+- 0.6 | 89.1+- 0.7 | 49.8+- 2.1 |
| RN50 (ImageNet-21k) | 97.7+- 1.3 | 76.9+- 0.5 | 86.8+- 0.7 | 50.1+- 0.8 |
| RN101 (ImageNet-21k) | 98.3+- 0.0 | 78.9+- 1.1 | 87.8+- 0.7 | 49.3+- 1.0 |
| DenseNet (ImageNet-1k) | 96.8+- 3.1 | 81.7+- 2.0 | 90.1+- 1.3 | 59.6+- 1.3 |

Table 8: $R^2$-score on CelebGlow

| modification models | random | composition | interpolation | extrapolation |
|---|---|---|---|---|
| BetaVAE | 79.4± 1.1 | -6.2± 2.5 | 10.9± 8.9 | -9.9± 6.3 |
| Ada-GVAE | 64.5± 0.8 | -3.3± 3.0 | 9.5± 5.7 | -3.6± 9.2 |
| SlowVAE | 89.0± 1.9 | -16.6± 11.3 | -10.9± 8.5 | -31.5± 15.2 |
| PCL | 95.8± 0.7 | 10.7± 10.2 | 21.8± 7.5 | -4.1± 10.3 |
| MLP | 97.0± 0.5 | 3.5± 4.0 | -37.5± 7.5 | -37.5± 12.1 |
| CNN | 99.8± 0.0 | 34.7± 1.4 | 26.3± 11.3 | 18.3± 10.0 |
| CoordConv | 98.6± 0.5 | 27.7± 19.3 | 18.5± 19.0 | 15.3± 27.5 |
| Coordinate Based | 93.5± 0.6 | 19.5± 5.3 | -50.5± 48.0 | -421.1± 286.5 |
| Rotation-EQ | 95.3± 0.6 | 23.6± 0.8 | 12.3± 12.1 | -44.3± 15.3 |
| Rotation-EQ-big | 99.9± 0.0 | 45.9± 1.0 | 45.8± 3.6 | 10.5± 2.8 |
| Spatial Transformer | 99.8± 0.0 | 16.1± 2.4 | 31.5± 12.2 | 9.0± 8.8 |
| RN50 | 100.0± 0.0 | 26.3± 1.5 | 29.4± 5.8 | 22.0± 5.3 |
| RN101 | 100.0± 0.0 | 26.1± 4.6 | 23.3± 15.7 | 20.7± 4.4 |
| DenseNet | 100.0± 0.0 | 44.0± 1.0 | 54.6± 3.3 | 7.0± 11.2 |
| RN50 (ImageNet-21k) | 99.8± 0.0 | 23.8± 3.3 | 43.6± 6.5 | 54.1± 1.9 |
| RN101 (ImageNet-21k) | 99.8± 0.1 | 37.0± 3.4 | 35.4± 15.4 | 41.6± 8.5 |
| DenseNet (ImageNet-1k) | 44.0± 79.0 | 49.0± 0.7 | 72.2± 3.4 | 38.9± 1.9 |

Table 9: $R^2$-score on MPI3D

| modification models | random | composition | interpolation | extrapolation |
|---|---|---|---|---|
| BetaVAE | 79.9+- 0.4 | 7.1+- 6.1 | 6.9+- 2.7 | 1.1+- 11.4 |
| Ada-GVAE | 57.6+- 1.2 | 1.2+- 4.0 | -22.4+- 2.5 | 5.0+- 4.2 |
| SlowVAE | 71.8+- 1.3 | -5.6+- 1.7 | 5.9+- 0.2 | -5.0+- 1.8 |
| PCL | 97.4+- 1.3 | 4.3+- 1.8 | 12.7+- 2.1 | -11.3+- 5.9 |
| MLP | 90.5+- 0.6 | 3.2+- 0.0 | -33.2+- 0.8 | -31.1+- 4.4 |
| CNN | 99.5+- 0.0 | 23.0+- 0.7 | 18.0+- 7.0 | -26.0+- 6.4 |
| CoordConv | 97.4+- 1.8 | 41.2+- 4.4 | 30.9+- 33.4 | 2.8+- 38.8 |
| Coordinate Based | 91.7+- 2.6 | 9.6+- 0.1 | -113.1+- 45.4 | -76.7+- 25.2 |
| Rotation-EQ | 94.8+- 0.7 | 22.6+- 4.9 | 15.5+- 6.6 | -38.1+- 9.5 |
| Rotation-EQ-big | 99.9+- 0.0 | 47.6+- 1.4 | 45.7+- 4.8 | -4.5+- 13.2 |
| Spatial Transformer | 99.6+- 0.1 | 23.6+- 6.4 | 37.7+- 18.4 | -2.2+- 0.6 |
| RN50 | 99.9+- 0.1 | 23.4+- 1.5 | 40.3+- 1.3 | 14.9+- 10.2 |
| RN101 | 99.9+- 0.1 | 19.7+- 1.0 | 37.8+- 3.7 | 9.8+- 11.7 |
| DenseNet | 99.9+- 0.0 | 33.2+- 0.5 | 51.4+- 3.0 | 4.0+- 6.5 |
| RN50 (ImageNet-21k) | 99.6+- 0.1 | 27.7+- 1.1 | 48.7+- 1.2 | 11.2+- 16.2 |
| RN101 (ImageNet-21k) | 99.8+- 0.1 | 39.2+- 3.7 | 39.2+- 7.6 | 5.6+- 10.5 |
| DenseNet (ImageNet-1k) | 62.9+- 51.9 | 41.8+- 2.6 | 73.7+- 4.9 | 34.3+- 0.5 |

Table 10: $R^2$-score on MPI3D-Toy

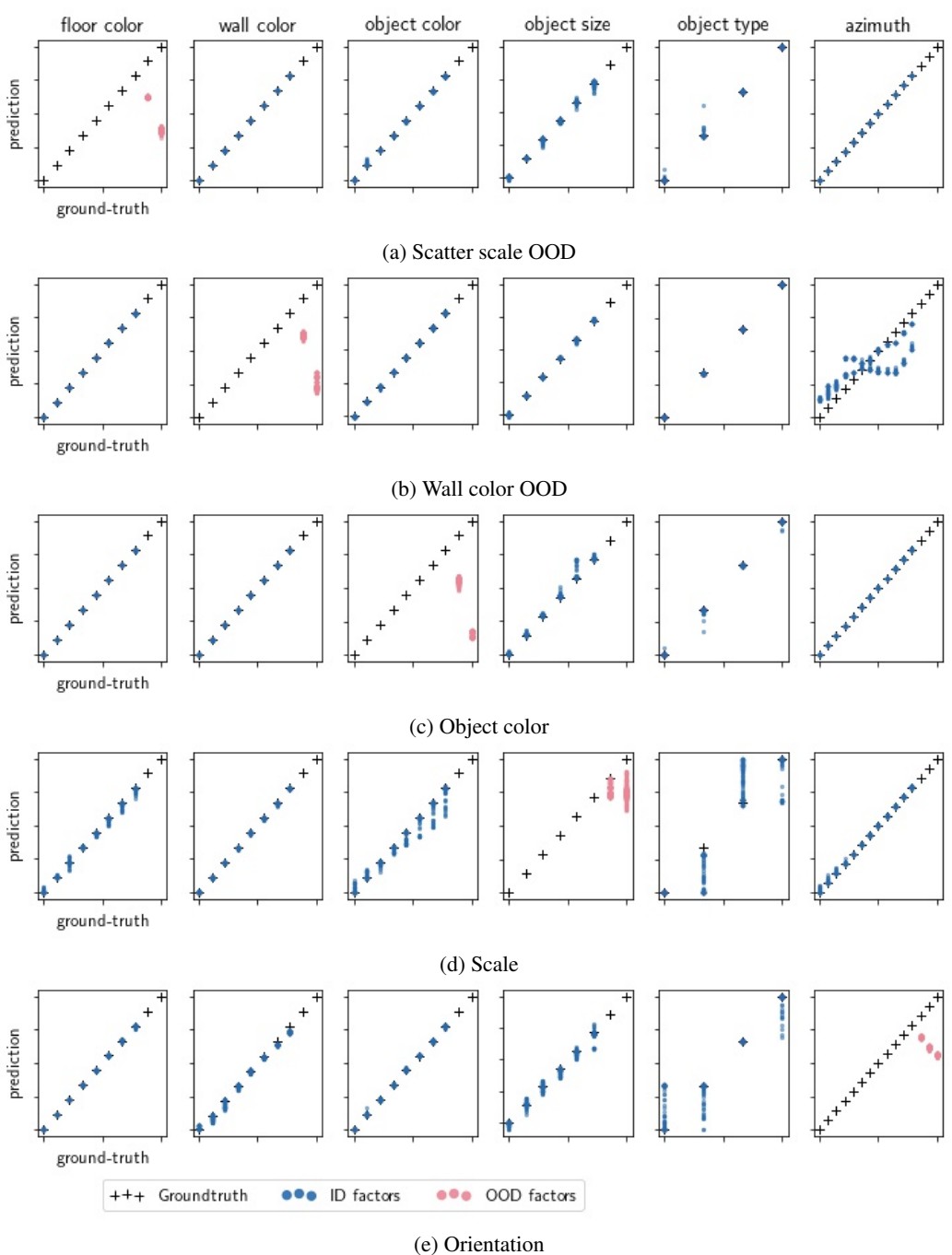

Figure 15: **Shapes3D extrapolation.** We show the qualitative extrapolation of a CNN model. The shape category is excluded because no order is clear.

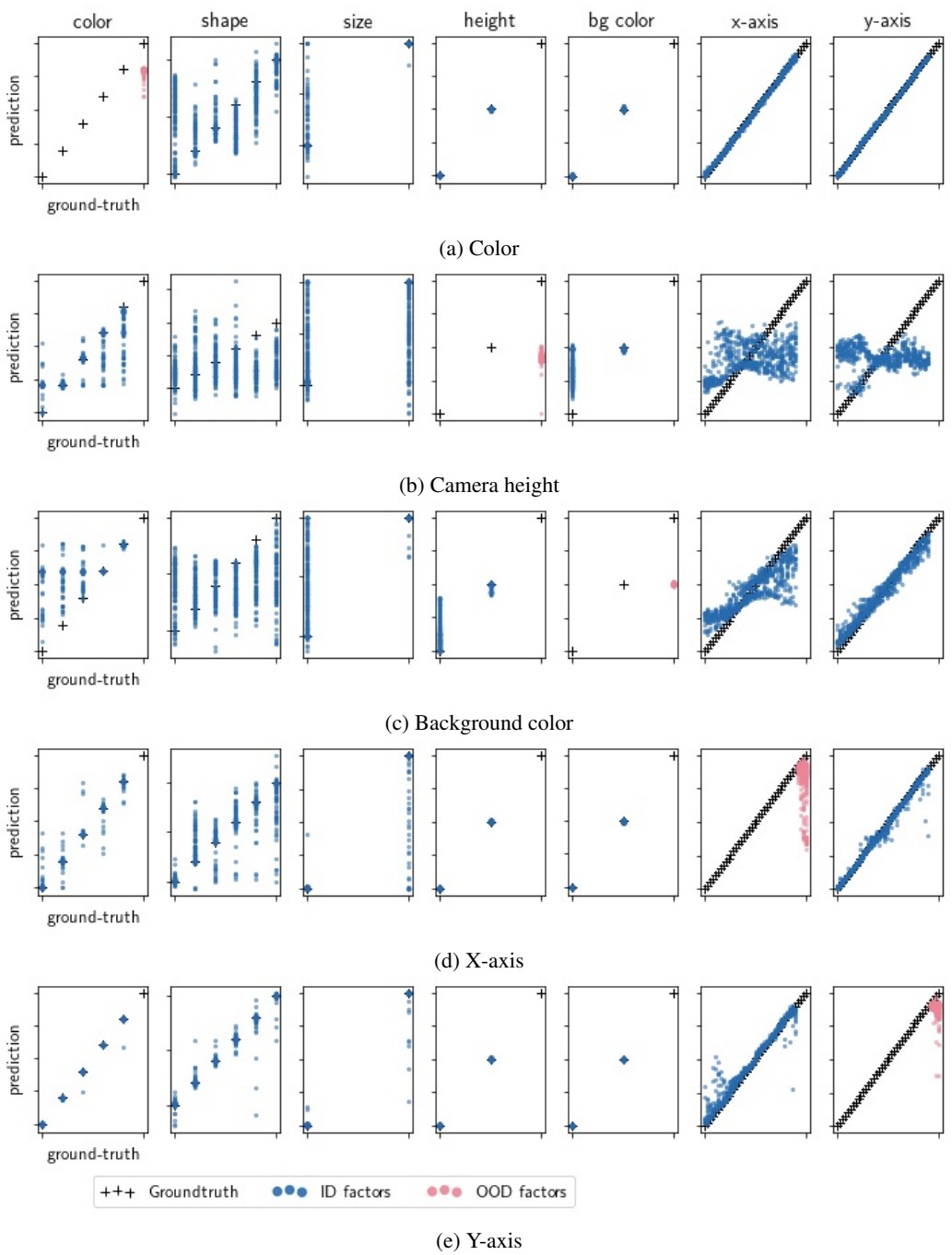

Figure 16: **MPI3D-Real extrapolation.** We show the qualitative extrapolation of a CNN model. The shape category is excluded because no order is clear. Size is excluded because only two values are available.

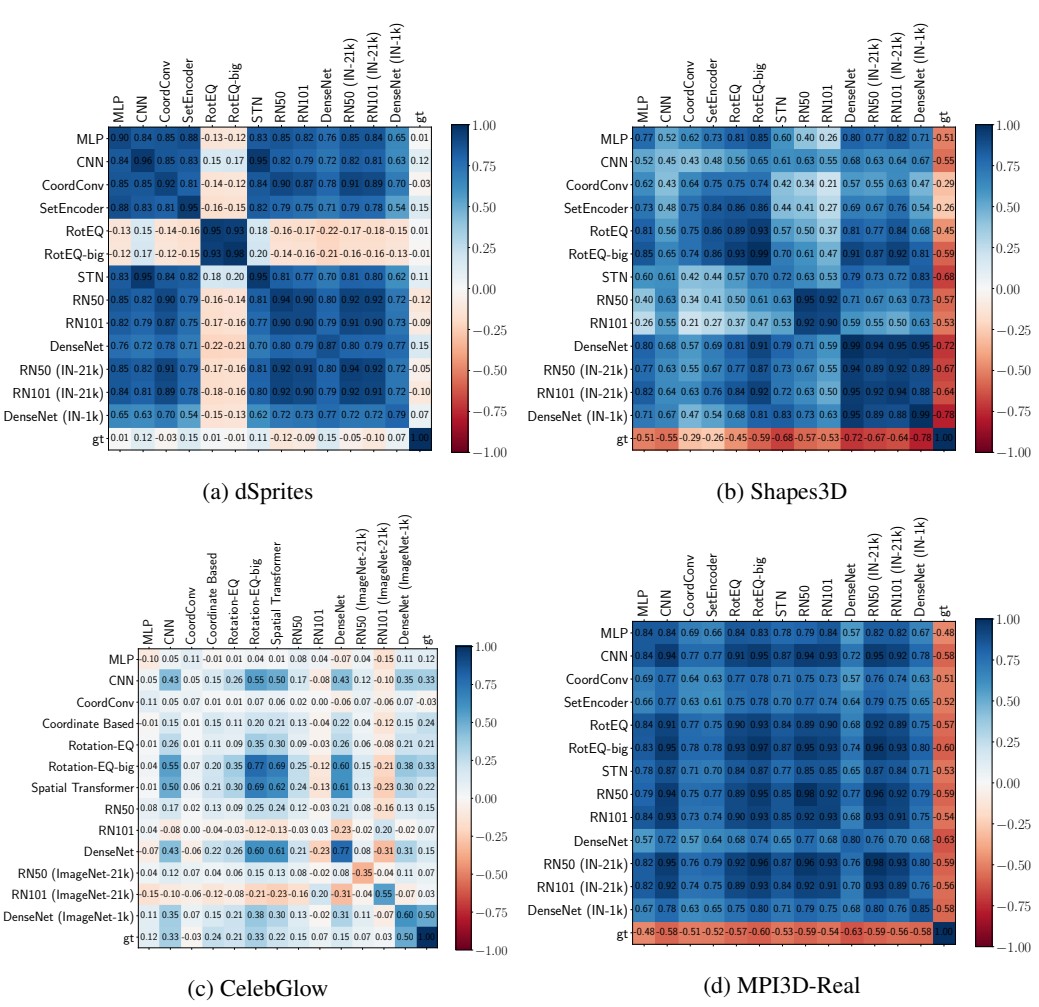

Figure 17: **Model similarity on extrapolation errors.**

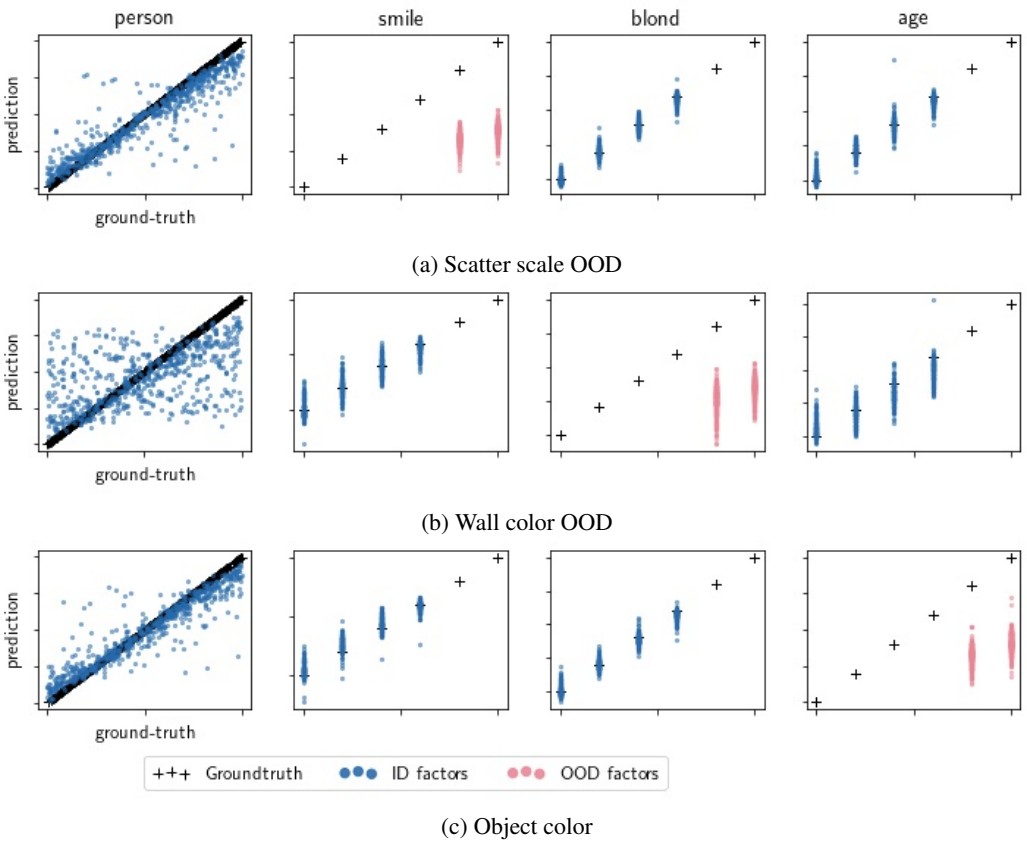

Figure 18: **CelebGlow extrapolation.** We show the qualitative extrapolation of a DenseNet (ImageNet 1-K) model. This corresponds, to the model with the highest correlation with the ground truth in Fig. 17 on the extrapolated factors (OOd factors). The person category is not extrapolated and used to measure correlations because no order is apparent.

