# OpenReview forum: "Visual Representation Learning Does Not Generalize Strongly Within the Same Domain"
_ICLR.cc/2022/Conference — ICLR 2022 Poster_

### Official Review · Reviewer_Yd4H · 2021-10-27

**Correctness:** 4
**Technical Novelty And Significance:** 3
**Empirical Novelty And Significance:** 3
**Recommendation:** 8
**Confidence:** 5

**Details Of Ethics Concerns:**

No ethics concerns

**Main Review:**

This is a very well-written and clear paper with sufficient implementation details for reproducibility.
Generalization in deep learning models remains an interesting question to study. As this study demonstrates, the problem is far from solved.
The paper surveys a wide range of 17 approaches on four datasets with increasing complexity of visual stimuli. However, I have concerns with the following issues, which I believe, limit the contribution of this paper.

1. At several places noted by the authors, also from Fig 4 and 5 (CelebGlow is better than MPI3D), there seems to exist variances in the generalization ability of models across different datasets or even within individual factors within each dataset. In addition to reporting the overall empirical results in entire datasets, can the authors provide explanations about why models perform better in certain factors/datasets and not the rest? And why would different variation factors within the same dataset result in different generalization ability?

2. In some cases, it is fair to say that deep models fail at some generalization tests. However, have the authors considered designing human experiments and quantitatively verifying human generalization abilities in similar problem settings? Human performance can serve as an essential benchmark to compare with the generalization ability of computational models. This is not a requirement (and certainly not the focus) of the current study. But it is hard to come up with an upper bound or even some sense of what level of generalization one should aspire to achieve.

3. The conclusions drawn from the paper are interesting and informative. However, there is a lack of explanations or insights about why these models fail to generalize in certain conditions. And if so, what are potential solutions to improve generalization ability.

4. The authors have focused on deep learning-based approaches. It is unclear that models without learning via backprop would be able to generalize. In other words, would other learning algorithms using Hebbian rules have better generalization ability (e.g. Boltzmann machines)?

5. The variation factors include translation, rotation, scaling, color, age, and so on. Have authors considered other dimensions? These factors are predefined and manually picked. The models might generalize better in some of these factors compared with the rest of factors. Have the authors considered ways to automatically exploit other factors that the models would show impairment in generalization abilities?


**Summary Of The Paper:**

The paper tests 17 unsupervised, weakly supervised, and fully supervised representation learning approaches to infer the generative factors of variation across three simple datasets in well-controlled conditions. In addition, the authors introduce a CelebGlow dataset, which is more complex. The generalization abilities are characterized as composition, interpolation, and extrapolation. The conclusions from these empirical observations on the experimental results are interesting and suggest that most networks fail to generalize.



**Summary Of The Review:**

The study poignantly points out failures of current models in generalization through a large number of empirical studies. The work is well written and clear. It is not particularly novel to argue that models fail to generalize, but the current study provides quantitative and extensive benchmarks in well-controlled settings. Beyond pointing to limitations of current models, the study does not offer new paths towards new algorithms or solutions to those problems. However, sometimes, understanding what the challenges are is a good step towards generating momentum to find better solutions.

---

> ### Author Response · Authors · 2021-11-16
> **Performance variations on datasets, humans, future improvements, other factors**
>
> We thank the reviewer for the valuable feedback.
>
> **Q1 Explain different performances across datasets and factors (review point 1.):**
> It is difficult to attribute the performance variations across datasets, as they are quite different. The datasets have different factor ranges, values and realizations in the image.
> To explain the high performances on Shapes3D on interpolation and composition, we hypothesize that this is due to the simplicity of the dataset itself. The underlying factors are mostly color-changes that could locally be generalized based on the used ReLU activation functions and their partial linear nature. Also, the dataset itself has a straight-forward spatial composition, as there is almost no variation over position. It is sufficient to look at a few pixels (middle, bottom, left/right) to get the colors of object, wall, and floor. This may be part of the reason why Shapes3D is so easy (see section 5.1 for further discussion on this).
>
> We further analyzed the suggested correspondence between the performance on individual factors. Here, we constrained the extrapolation splits such that only a single factor is out-of-distribution and measured the $R^2$ -squared score. The results are visualized over all models https://ibb.co/jykN84s (Note that this is similar to Fig. 5 but broken down to individual factors). While we do observe variations depending on individual factors, it is hard to draw systematic conclusions from this.
>
>
> **Q2 Human performances on quantifying generalization (review point 2.):**
> As pointed out by the reviewer, this is beyond the scope of the paper. It is also beyond the expertise of the authors. Nonetheless, this is philosophically indeed interesting as some concepts are easy to generalize and some are not. E.g., concepts like spatial dimensions are very hard to generalize for humans even though 1D (a string), 2D (a sheet of paper) and 3D objects (our world) are simple examples.
> Furthermore, simple counting itself is not necessarily extrapolated. For instance, 139 Aboriginal Australian languages have an upper limit at “three” or “four” and use “several” or “many” to refer to larger quantities [1*, 2*].
> For other studies that allow for a comparison with humans, we refer to our related work section, such as IQ-tests for neural networks [3*], a general setup on comparing humans and machines [4*,5*].
>
> [1*] Barras, C. (2021).  How did neanderthals and other ancient humans learn to count?Nature,594(7861),22–25.
> [2*] Bowern, C. and Zentz, J. (2012). Diversity in the numeral systems of australian languages.AnthropologicalLinguistics,54(2), 133–160.
> [3*] David Barrett, Felix Hill, Adam Santoro, Ari Morcos, and Timothy Lillicrap. Measuring abstract reasoning in neural networks. In International conference on machine learning, pp. 511–520. PMLR, 2018.
> [4*] C. M. Funke, J. Borowski, K. Stosio, W. Brendel, T. S. A. Wallis, and M. Bethge. Five points to check when comparing visual perception in humans and machines. Journal of Vision, Mar 2021. URL https://jov.arvojournals.org/article.aspx?articleid=2772393.
> [5*] B. Peters, N. Kriegeskorte. Capturing the objects of vision with neural networks. Nature human behaviour https://www.nature.com/articles/s41562-021-01194-6
>
>
> **Q3 Why do models fail and potential improvements (review point 3.)?**
> We hypothesize that the current inductive biases for models are not strong enough to facilitate learning towards more principled solutions.
> Please refer to Q4 of reviewer Cjpj for future ideas.
>
> **Q4 Other learning algorithms (Hebbian, Boltzmann machines, non-backprop) (review point 4.):**
> For common classical approaches, it is straightforward to show that they will not extrapolate. E.g., a nearest neighbor regression or a Random Forest regressor with standard deviation reduction only predict values in previously observed ranges. Other approaches are difficult to implement as we are dealing with image data (64x64x3). Thus, we have not considered other algorithms.
>
> **Q5 Other factors than the ones considered? Automize the process of detecting factors along models show impaired generalization abilities (review point 5.):**
> This is an interesting idea for future research. Especially, such automated detection methods could allow for an identification of difficult factors in the real world. Subsequently, the research community could focus on those. In our standpoint, some preliminary works on unsupervised and identifiable disentanglement could be a promising direction for this.
>
>
> We hope we were able to address your concerns. Please let us know if further clarifications are needed.

---

> > ### Comment · Reviewer_Yd4H · 2021-11-24
> > **Thanks for the answers**
> >
> > Thank you for all the answers, which are great.
> > I am not sure about the connection to aboriginal australian languages... I meant something simpler, i.e., benchmarking human performance on the same tasks.
> > The broken down version of Fig. 5 is quite interesting. Thanks for including this.

---

### Official Review · Reviewer_Cjpj · 2021-11-01

**Correctness:** 4
**Technical Novelty And Significance:** 4
**Empirical Novelty And Significance:** 3
**Recommendation:** 8
**Confidence:** 3

**Main Review:**

Positives:
- The paper is extremely well motivated and addresses a very important problem in the field of generalization in NNs
- The dataset also looks very promising
- The experiments are extremely well done and thorough
- The idea of having "conclusion" section in each subsection in the experiments is a very nice idea.

Negatives:
- I would prefer if the figures had appropriate captions - all tables and figure captions should be self-sufficient. There should be sufficient details in them such that I am able to understand what the paper is doing.
- I am slightly concerned about what "new" information this adds to the field - a lot of the conclusions in this paper were well known in the community, to my knowledge. This does not discredit the work at all, since it is always good to have experimental support related to "intuitions" that people have had, but I would be keen on the authors to answer what "previously unconfirmed" things they really discovered in this paper.
- the paper states "Instead of extrapolating, all models regress the OOD factor towards the mean in the training set." which should be clarified as I am not sure what that means. The ratio of distances idea makes sense, but I am not sure what the authors are trying to conclude with that statement.
- Is there a reason why CelebGlow outperforms MPI3d and shapes3d considerably in Figure 4 and 5 on the "extrapolation" benchmark?

Limitation:
- The paper adaquately points out where and how the NNs fail to adapt, without giving a clear direction of what are possible ways to remediate the current gap. It would be a useful discussion point in the paper.

**Summary Of The Paper:**

The paper introduces a new dataset: CelebGLOW which is a controllable environment generation dataset, which can be used in the same form as significantly less complex datasets, such as sprites. The paper evaluates over 3 key inductive biases using the dataset: "representational format" (using images in this paper), neural network architectural variants (MLPs/CNNs/ transformers, etc), and ability to perform transfer learning. The paper evalutes the models on 4 given modes of generalization: interpolation, extrapolation, compositional learning and "random", and finds extremely fascinating conclusions through extensive experimentation.

**Summary Of The Review:**

It is a well-written paper with great analysis and a very useful study to have. I have small concerns about the novelty of the conclusions, but the paper is still a great addition to the community.

---

> ### Author Response · Authors · 2021-11-16
> **Captions; new findings, conclusions from extrapolation, future work**
>
> We thank the reviewer for their positive and motivating feedback.
>
> **Q1 Self sufficient captions:**
> We fully agree and will revise the paper from this perspective.
>
> **Q2 What are "previously unconfirmed" things discovered in this paper?**
> While it has previously been shown that neural networks for representation learning do not generalize well out-of-distribution, we show that this is even true in a simpler setup: Even if the factors are present in the data, representation learning neural networks do not seem to generalize strongly along them.
> Furthermore, provably identifiable methods that may naively be expected to capture the true mechanism (at least within the training support) struggle with OOD generalization.
> We further provide novel insights on “how” neural networks generalize by showing that a diverse set of different inductive biases extrapolate quite similar.
>
> **Q3 Conclusions from extrapolation towards the mean unclear:**
> We first show that the extrapolation behavior across models is quite similar. We further show that extrapolation towards the mean is a prototypical behavior. As a conclusion, for further research, we hope to foster new approaches with a different extrapolation behavior on out-of-distribution data.
>
>
> **Q4 The paper does not provide a discussion “of what are possible ways to remediate the current gap”:**
> Here, some speculative ideas for future works:
> * Train one architecture per factor, similar to [*1].
> * Equivariant layers for shift, rotation and scale could serve as a basis to solve dSprites. Most equivariant model implementations include equivariances in additional channels. Here, we could additionally hard-code a mapping to infer a single factor. For instance, to infer the categorical variable “shape” on dSprites, a pooling operation over the equivariant channels for shape in the last layer could be sufficient. However, this is dissatisfying because the solution is hard-coded and would not transfer to, e.g., a rotation in 3D projected onto 2D because in this case, some symmetry assumption about occluded parts of the objects have to be made.
> * EQN networks [*2] with suiting activation functions (e.g., cosine for rotations or Linear for shifts) with Coordinate based SetEncoders.
> * Use a set of basis functions to transfer to a space that fosters generalization.
>
> [*1] Spandan Madan, Timothy Henry, Jamell Dozier, Helen Ho, Nishchal Bhandari, Tomotake Sasaki, Frédo Durand, Hanspeter Pfister, Xavier Boix. When and how do CNNs generalize to out-of-distribution category-viewpoint combinations? arXiv preprint arXiv:2007.08032 [cs.CV].
>
> [*2] Martius, G., & Lampert, C. H. (2016). Extrapolation and learning equations. arXiv preprint arXiv:1610.02995.
> We hope, we were able to address your concerns. Please let us know if further clarifications are needed.

---

### Official Review · Reviewer_ZKCQ · 2021-11-03

**Correctness:** 4
**Technical Novelty And Significance:** 2
**Empirical Novelty And Significance:** 3
**Recommendation:** 8
**Confidence:** 4

**Main Review:**

I think this is a very good paper. The writing is exceptionally clear and the problem framing and literature review will be an especially valuable resource for future work. The experiments are well designed and thorough, and the findings are valuable contributions to an important area.

The main weakness I see is that I'm not sure there is a lot of information gain from this paper. The results are more or less what I think most readers will expect. The findings seem in line with past work on generalization and with conventional wisdom -- ID works, OOD works worse, inductive biases can help on OOD, etc. That's not to say the present work is not valuable -- there are open questions here and the present paper is one of the most extensive studies I have seen on them. Just that the paper doesn't provide entirely unexpected answers. Because of this, I think the most valuable contribution of the paper may be the benchmark it provides, on top of which future studies may find something really new.

To elaborate further, the related work covers numerous papers that have come to roughly similar conclusions. Two more papers come to mind that also have similar conclusion but were not discussed:
* Packer et al., “Assessing Generalization in Deep Reinforcement Learning”, 2019
-- This paper also studied ID vs OOD generalization but in the context of reinforcement learning. Figure 1 from Packer et al. shows their train/test setup, which is quite related to the current paper's settings in Figure 2. The conclusions are similar to the current paper: 1) extrapolation is harder than interpolation, 2) SOTA algorithms that are supposed to "solve" this problem fail.
* Jahanian et al., "On the 'Steerability' of Generative Adversarial Networks", 2020
-- This paper studied the ability of GANs to extrapolate in their latent space. The finding is that they struggle to generate transformations that extend beyond the distribution seen during training. These results are similar to the findings in the current paper on the failure of VAE latent representations to extrapolate beyond the training data.

This is all to say I think there is ample prior literature that make the present conclusions unsurprising. But thorough work on this topic, and new benchmarks, is still valuable and that's what the current paper provides. I should also note that the finding about modularity is something I hadn't seen before, and I think that's a valuable contribution as well.

Aside from this, I think the paper is very solid. A few minor comments follow:
1. Using CelebGlow feels a bit awkward since it is a generative model fit to data, and then you are again fitting samples from this model with another generative model. I wonder if there could be some bias where the samples are easier to model with a VAE since they were generated with a related model (Glow)... It's probably all fine but some commentary on this could be useful.
2. Repeated reference to Hendrycks and Diettrich 2019.
3. I would say inductive biases 1 and 3 overlap: inductive bias 1, as it is implemented in the paper, could be considered a special case of transfer learning where the pretraining is done with VAEs. This could be clarified to avoid implying that these are independent inductive biases.
4. “if factors are located in a particular edge of the FoV hyper cube given by all FoVs” — “edge” —> “corner”?
5. “Here, we further see that, on average, the performances seem to increase as we increase the supervision signal.” — This is a bit vague I don’t see it fully reflected in the figure. This point could be made more precise. What does “increase the supervision signal” refer to?
6. “We find that the degree of downstream performance correlates weakly but positively with the degree of disentanglement (Pearson ρ = 0.63, Spearman ρ = 0.67)” — I’m not sure I would call these weak correlations. In many fields I believe this would be considered a strong correlation.
7. “Existing notions of disentanglement models with a readout MLP do not help to facilitate the learning of the underlying mechanisms in the tested datasets.” — I don’t understand this conclusion. The correlations are substantial. My understanding of the results is that greater disentanglement _does_ correlate with better ability to identify the underlying mechanisms. Appendix Fig 7 seems to support this as well, with all but one of the correlations being positive.


**Summary Of The Paper:**

This paper presents an empirical study of generalization in visual representation learning. The paper compares in-distribution (ID) generalization to out-of-distribution (OOD) generalization of three types -- interpolation, extrapolation, and composition. Datasets are constructed with several factors of variation where the training split exhibits some factors and the test split exhibits others, constructed to test the ID and OOD settings. A variety of different representation learning models with different kinds of inductive bias are tested. The main findings are that: 1) all models perform better ID than OOD, indicating that all methods fail to find the true underlying generative mechanisms that created the data, 2) when some factors are ID and others are OOD the ID factors are modeled well despite that the OOD factors may be modeled poorly, demonstrating a kind of modularity between the learning of different factors. The paper will be accompanied by a benchmark for others to improve on this task.

**Summary Of The Review:**

This is a solid paper that contributes new empirical findings and a new benchmark on an important topic. I believe this work will stimulate future work on representation learning, disentanglement, and OOD generalization. I don't see any major errors and recommend accepting.

---

> ### Author Response · Authors · 2021-11-16
> **Positioning of results; related work; minor points**
>
> We thank the reviewer for their positive and motivating feedback.
>
> **Q1 Results unsurprising:**
> We agree that a multitude of publications show that neural networks do not generalize out-of-distribution. However, we try to convey a different and more fine-grained message: even if the factors are present in the data, strong generalization is still limited. Thus, this simpler problem should be an intermediate milestone on the path towards more general robustness (e.g. to factors that are not present or very rare in the training data). Furthermore, we provide novel insights on “how” neural networks generalize by showing that a diverse set of different inductive biases extrapolate quite similar.
>
>
> **Q2 Additional related work:**
> Thanks for the pointers, we are happy to add and discuss the work of [1*] and [2*] in our related work.
>
> [1*] Packer et al., “Assessing Generalization in Deep Reinforcement Learning”, 2019
> [2*] Jahanian et al., "On the 'Steerability' of Generative Adversarial Networks", 2020
>
>
> **Minor points:**
> **Q3 Add clarification on CelebGlow setup feels a bit awkward [circular setup] (review point 1.):**
> This was mostly done out of the lack of existence (to the best knowledge of the authors) of a natural dataset that allows for a full control of factors of variations for our splits. We will clarify this in the motivation section describing the dataset.
>
>
> **Q4 Overlap of inductive biases 1,3 (review point 3.):**
> We agree, as disentanglement and other representation learning methods are common pre-training methods. We will add the suggested overlap to the manuscript,
>
>
> **Q5 Clarification on “‘performances seem to increase as we increase the supervision signal’”  (review point 5.):**
> What we meant: On the MPI3D-Real dataset the R-squared performance increases for the “Transfer” models that have been trained on additional data compared to their architectural twins that have only been trained on the dataset itself (RN50, RN101, DenseNet). We’ll rephrase this more accurately in the manuscript.
>
>
> **Q6 Correlations on downstream performance and disentanglement, actually more positive than emphasized (review point 6./7.):**
> We think the connection sounds more negative than anticipated.
> We will rephrase the correlation from “low” to “medium” and also highlight the experiments in the appendix. Here, a perfect disentanglement automatically leads to perfect downstream performance (Appendix A).
>
>
> **Other points:**
> Thanks for the other minor suggestions. For brevity, we’ll incorporate them directly into the manuscript (semantics, references, …)
>
>
>
> We hope, we were able to address your concerns. Please let us know if further clarifications are needed.

---

> > ### Comment · Reviewer_ZKCQ · 2021-11-29
> > **thanks for the thorough responses; maintaining positive rating of paper**
> >
> > Thanks for the thorough responses to my comments and those of the other reviewers.
> >
> > I'm keeping my rating at 8 -- I think this is a solid paper that should be accepted.
> >
> > On the "unsurprising results": I agree that the current message is more fine-grained than much of the prior work on generalization, but I think even this fine-grained point has precedents. For example, Packer et al. does specically test the setting where all factors are present in the training data, yet generalization along these factors fails. I think the current messaging in the paper is appropriate, and I appreciate the discussion of additional related work, but I would be careful not to overemphasize the novelty of this fine-grained question/finding.

---

### Official Review · Reviewer_g7qs · 2021-11-08

**Correctness:** 2
**Technical Novelty And Significance:** 2
**Empirical Novelty And Significance:** 2
**Recommendation:** 8
**Confidence:** 3

**Main Review:**

The direction of this work is nice and the problem tackled by this paper is indeed important. The obtained results are nice and there is indeed some potential value in this work. However, I have a few concerns with the paper:

i) To test generalizability of neural networks, test datasets are constructed using four methods: interpolation, extrapolation, random, and composition. However, previous works uses these approaches as data augmentation to improve the generalizability of neural networks.  For example, multi-scale inputs during training in YOLOv2 allows it to predict well across different input dimensions. Another example is random erasing, which also improves generalization ability of neural networks. A discussion about these methods to improve generalization ability is missing. Also, what happens to the generalizability of neural networks when they are trained with these previous approaches?

YOLOv2: https://arxiv.org/pdf/1612.08242.pdf

Random erasing: https://arxiv.org/pdf/1708.04896.pdf

CutMix: https://arxiv.org/abs/1905.04899

Random Augmentation: https://arxiv.org/abs/1909.13719

ii) The generalization performance of neural networks drops when we move from artificial datasets to real-world datasets. However, the datasets studied in this paper are simple and limited, and far from real-world datasets. They do not consider many factors that are natural in real-world datasets, including non-rigid deformations and intraclass variability (e.g., people wear different types of glasses).

iii) Previous works have shown that larger models generalize better than smaller models. I do not see any argument about the model capacity and generalizability in the paper. Can authors comment on that?

https://arxiv.org/pdf/1802.08760.pdf

=========

Clarification:

In page 4, it is stated that "Combined with a set operation such as pooling, CNNs then achieve translation invariance." To my understanding, translation invariance is one of the fundamental properties of convolutions. I think operations like pooling forces CNNs to learn multi-scale representations

https://www.ijcai.org/Proceedings/83-2/Papers/091.pdf


**Summary Of The Paper:**

This paper studies the generalization ability of neural networks in out-of-distribution (OOD) settings on simple datasets (such as Shapes3D) and shows that generalization performance drops when real-world datasets are used in comparison to artificial datasets.

**Summary Of The Review:**

This paper studies generalizability of neural networks using four factors (interpolation, extrapolation, random, and composition). Previous works on better data augmentation policies uses similar factors to improve the generalization of visual recognition networks. So, the results of this paper are not surprising.


=================
Post rebuttal: I am convinced with author's response and I am leaning towards acceptance.

---

> ### Author Response · Authors · 2021-11-16
> **Data augmentation; positioning of datasets; capacity vs generalization; clarification**
>
> **Q1 Data augmentation methods (point i in review):**
> Thanks for raising this interesting point. We followed your suggestions and ran several experiments incorporating data augmentation techniques.
>
> As data augmentations we applied random erasing, Gaussian Noise, small shearings, blurring, …
> Note that, we could not use all augmentations. For instance, shift augmentations would lead to ambiguities with the “shift” factor in dSprites. Randomly sampled training augmentations for all datasets are shown here: https://ibb.co/09fL0DT
>
> Next, we trained CNNs with and without data augmentations on all four datasets (dSprites, Shapes3D, MPI3D, CelebGlow) on the extrapolation splits with multiple random seeds.
>
> The results are visualized here https://ibb.co/qCQwpdQ . For the mean $R^2$-performance, we observe no significant improvement by adding augmentations. However, the overall spread of the scores seems to decrease given augmentations on some datasets. We, explain this by the fact, that the augmentations enforce certain invariances narrowing the solution space of optimal training solutions by providing a further specification (specification in the sense of [1*]). We will add this study to the appendix.
>
> [1*] D'Amour et al., “Underspecification Presents Challenges for Credibility in Modern Machine Learning”.
>
>
> **Q2 Considered datasets far from real-world datasets (non-rigid transformations, wearing glasses, …) (point ii in review):**
> We agree that our benchmark does not cover the variability of the real world. Our goal is to introduce a benchmark that is positioned between standard datasets that have similar test and train distributions (e.g., MNIST test, train splits) and datasets with very complex test-time perturbations, such as ImageNet-C [1*] or ImageNet-R [2*]. In our datasets, factors are out-of-distribution but have to be partially observed during training. Thus, our dataset can be seen as an important intermediate milestone towards general robustness. Furthermore, as we fix the underlying generative model, our setup should foster more principled solutions relying on learning underlying mechanisms of the data.
>
> Moreover, we would like to emphasize that our factors like “smiling” or “age” on our introduced the CelebGlow dataset are non-rigid. In principle, the “glasses” dimension is also present in glow latent space [3*]. We are currently looking whether this could be included into the dataset.
>
> [1*] https://github.com/hendrycks/robustness
> [2*] https://github.com/hendrycks/imagenet-r
> [3*] https://github.com/openai/glow/blob/master/demo/model.py#L219 (Eyeglasses)
>
>
> **Q3 Connection of capacity generalization behavior (point iii in review):**
> In the appendix C, “Hyperparameter Tuning Ablation”, we tried a small 6 layer and a larger 9 layer convolutional networks (Fig. 9). Here, we observed no significant improvements. Similarly, when comparing the $R^2$-scores for the ResNet50 and ResNet101 (Fig. 4 and 11), we observe no clear trend. Lastly, also the Rotation-Equivariant networks are implemented with two widths. Again, we observed no clear connection between performance and model size in our benchmark.
> Nevertheless, in principle all models should have sufficient capacity, as almost all models can achieve high $R^2$-scores on the Random split.
>
>
> **Q4 clarification on convolutions, pooling and invariance:**
> The standard convolutional operations in neural networks are shift-equivariant. E.g., a shift in the input leads to proportional changes in the output (up to edge effects). This is different from shift-invariance where a shift in the input would not change the output, see [1*, section 5 in 2*]. Normally, this change is implemented within the channel operations. Thus, when combining the equivariant convolutions with a set operation such as pooling over the channels, we achieve shift-invariance. We will add a footnote clarifying this in the paper.
>
> [1*] https://en.wikipedia.org/wiki/Convolutional_neural_network
> [2*] Cohen et al., “Group Equivariant Convolutional Networks”.
>
>
>
> We hope, we were able to address your concerns. Please let us know if further clarifications are needed.

---

> > ### Comment · Reviewer_g7qs · 2021-11-19
> > **Thanks**
> >
> > Thank you for your response and I really appreciate the additional results that are provided. It is an interesting observation that adding augmentation reduces the variance in scores, likely because these augmentations improve generalization performance (as illustrated in previous works).
> >
> > 6- and 9-layer neural networks are small and I think they have similar number of parameters. So, it is hard to make any conclusions with such networks. Irrespective of that, model parameters should be included in the paper.

---

### Author Response · Authors · 2021-11-22
**Overview of changes**

We thank all reviewers for their valuable feedback.

Here, we provide a coarse summary of the changes to the paper. These changes and other points are addressed in more detail in the individual author responses.


**Updates:**
* Clarification on positioning of the paper/ novelty. While a myriad of publications show that neural networks do not generalize out-of-distribution, we try to convey a different and more fine-grained message: even if the factors are present in the data, strong generalization is still limited. This makes our experimental setup an important and attractive intermediate milestone on the path towards more general robustness. We further provide an in-depth analysis for various inductive biases and show that they behave surprisingly similar.
* Added experiments with data augmentations section in Appendix F (reviewer g7qs).
* Added and discussed references [Packer et al., 2019] and [Jahanian et al., 2020] (ZKCQ)
* Added more elaborate and self-consistent figure captions. Thus, the main figures alone should demonstrate the coarse setup and results (reviewer Cjpj).
* From our observation that models prototypically extrapolate towards the mean, we added the  conclusion that future models should explore models that behave differently (section 5.3) (reviwer Cjpj).
* We provide additional insight on how models perform when specific factors are out-of-distribution in appendix G (reviewer Yd4H)


**Minor:**
* Added  an example for equi- and invariance in section 3.2 (reviewer g7qs)
* Clarified motivation of CelebGlow dataset in appendix B (reviewer ZKCQ point 1.)
* Fixed double reference of Hendrycks et al. (ZKCQ point 2.)
* Implied connection between inductive biases for pretraining and representational format in section 3.3 (ZKCQ point 3.)
* Edge -> corner in section 4.2 (reviewer ZKCQ point 4.)
* Added clarification on effects of pretraining on MPI3D in section 5.1 (ZKCQ point 5.)
* Rephrased correlation between disentanglement more precisely in section 5.4 and section 6 (ZKCQ point 6/7)
* Moved some Shapes3D plots to the appendix for space.

---

### Decision · Program_Chairs · 2022-01-20

**Decision:**

Accept (Poster)

**Comment:**

This paper presents a through study of generalization in visual representation learning. It compares in distribution generalization to out of distribution generalization using a comprehensive benchmark. The paper received very positive reviews from all reviewers. Reviewers agreed that the paper has several strengths: It is very well written, the presented benchmark is very useful and the analysis is thorough. One concern that was brought up by the reviewers was that a majority of the presented findings are expected and in a sense, known to the community. The authors have addressed this concern by pointing out that their findings are more fine grained than past works and that their proposed benchmark is a stepping stone towards measuring general robustness. I must note that in spite of this concern, all reviewers have maintained their strong acceptance scores. I agree with the reviewers. This paper makes a strong contribution to this important problem via its benchmark and analysis, which future works can build off of, and hence I recommend acceptance.